# A systematic review of security and privacy issues in the internet of medical things; the role of machine learning approaches

Shilan S. Hameed[1,2], Wan Haslina Hassan[1], Liza Abdul Latiff[3] and Fahad Ghabban[4]

[1] Malaysia-Japan International Institute of Technology (MJIIT), Universiti Teknologi Malaysia, Kuala Lumpur, Malaysia
[2] Directorate of Information Technology, Koya University, Koya, Kurdistan Region, Iraq
[3] Razak Faculty of Technology and Informatics, Universiti Teknologi Malaysia, Kuala Lumpur, Malaysia
[4] Information Systems Department, College of Computer Sciences and Engineering, Taibah University, Medina, Saudi Arabia



Corresponding author
Shilan S. Hameed,
hameed.s@graduate.utm.my

## ABSTRACT

**Background:** The Internet of Medical Things (IoMTs) is gradually replacing the traditional healthcare system. However, little attention has been paid to their security requirements in the development of the IoMT devices and systems. One of the main reasons can be the difficulty of tuning conventional security solutions to the IoMT system. Machine Learning (ML) has been successfully employed in the attack detection and mitigation process. Advanced ML technique can also be a promising approach to address the existing and anticipated IoMT security and privacy issues. However, because of the existing challenges of IoMT system, it is imperative to know how these techniques can be effectively utilized to meet the security and privacy requirements without affecting the IoMT systems quality, services, and device's lifespan.

**Methodology:** This article is devoted to perform a Systematic Literature Review (SLR) on the security and privacy issues of IoMT and their solutions by ML techniques. The recent research papers disseminated between 2010 and 2020 are selected from multiple databases and a standardized SLR method is conducted. A total of 153 papers were reviewed and a critical analysis was conducted on the selected papers. Furthermore, this review study attempts to highlight the limitation of the current methods and aims to find possible solutions to them. Thus, a detailed analysis was carried out on the selected papers through focusing on their methods, advantages, limitations, the utilized tools, and data.

**Results:** It was observed that ML techniques have been significantly deployed for device and network layer security. Most of the current studies improved traditional metrics while ignored performance complexity metrics in their evaluations. Their studies environments and utilized data barely represent IoMT system. Therefore, conventional ML techniques may fail if metrics such as resource complexity and power usage are not considered.

## BACKGROUND

### IoMT and its classification

The Internet of Thing (IoT) is a fast growing technology by which the infrastructures, computerized machines, physical things, applications and individuals are allowed to connect, communicate, capture and exchange information through networking (*Farahani et al., 2018*; *Noor & Hassan, 2019*). Hence, the Internet of Medical Thing (IoMT) is the application of IoT in medicine and healthcare industry (*Alsubaei, Abuhussein & Shiva, 2019a*; *He et al., 2018*). It is anticipated that with the implementation of the IoMT a significant improvement in the efficiency and standard of treatment is achieved thanks to the steady innovations in IoT including the development in microprocessors, biosensor architecture and evolving 5G technologies (*Ahad, Tahir & Yau, 2019*). However, it is hard to maintain a specific architecture as a baseline due to the variety of devices and their different usage. Consequently, various approaches for the IoT architectures and layers were presented in literature (*Sethi & Sarangi, 2017*; *Weyrich & Ebert, 2015*) such as protocol-based architecture (*Burhan et al., 2018*; *Mosenia & Jha, 2016*; *Weyrich & Ebert, 2015*) and data processing stages known as edge, fog, and cloud based approaches (*Escamilla-Ambrosio et al., 2018*). In the current review, we present two common architectures of the IoMT. In the first architecture, the IoMT is treated as to be composed of three main elements: (i) network of wireless body sensors, which is device layer, (ii) internet connected smart access points (gateways), which is fog layer, and (iii) cloud computing and big data service, which is cloud layer (*Rahmani et al., 2018*). The second architecture is simply defined by sensing/device layer, network layer, and server (personal/medical) layer (*Sun, Lo & Lo, 2019*). However, some researchers have divided the IoMT architecture into more than three layers based on the requirements adopted for a specific application of the IoMT (*Elrawy, Awad & Hamed, 2018*; *Grammatikis, Sarigiannidis & Moscholios, 2019*). Furthermore, there are four types of smart medical devices in the IoMT that are categorized based on their location of implementation on human body (*Nanayakkara, Halgamuge & Syed, 2019*). Table 1 shows a brief explanation on the types of the IoMT devices and their properties. According to the U.S. Food and Drug Administration (FDA), the IoMT devices can be also classified based on their risk value. Implantable devices such as Implantable cardioverter-defibrillator (ICD), Electrocardiography (ECG), Electromyography (EMG), and Electroencephalography (EEG) are under high-risk category. Hence, they are regulated and certified by the FDA. However, non-implantable devices such as fitness trackers, and smart watches are under low-risk category. Therefore, they neither require certification nor regulation by the FDA (*Jaigirdar, Rudolph & Bain, 2019*).

Medical devices and biosensors are responsible for capturing body's vital signs, and transferring a huge real-time raw biological data such as heart rate, brain signal, temperature of the body, and glucose level in blood (*Dang et al., 2019*). These raw data are accumulated and processed at the personal servers that are either devices located near the patient's body such as mobile phones, medical programmers, and laptops or devices placed far from body such as gateways and routers. Additionally, personal servers usually

**Table 1 The types and characteristics of medical devices used in the IoMT system.**

| Device type | Placement | Example | Risk Value | References |
|---|---|---|---|---|
| Implantable | Within the human tissues | deep brain implants, heart pacemaker and insulin pump | High | *Santagati, Dave & Melodia (2020)* |
| Wearable | On the human body | smart watches, fitness devices | Low | *Tseng, Wu & Lai (2019)* |
| Ambient | Outside the human body | elderly monitoring devices in smart home | Low | *Pandey & Litoriya (2020)* |
| Stationary | Inside hospitals | medical image processing devices of MRI and CT-Scan | Low | *Nanayakkara, Halgamuge & Syed (2019)*, *Xu et al. (2019)* |

**Note:**
Each row represents a different type of medical device, and each column represents characteristics of that device.

include a computing analysis facility, which is coupled with a local archiving database in order to store the initial records of the patient. Furthermore, its warning system alerts the patient whenever an abnormality is observed (*Newaz et al., 2020*). The communication link between body area network (sensors attached to or close to human body) and personal server is usually employed by low-power wireless networking technologies such as Bluetooth or Bluetooth Low Energy (BLE), Near-Field Communication (NFC), Radio-Frequency IDentification (RFID), Zigbee and Z-Wave. Noteworthy, Bluetooth is mostly used in wearable devices, while RFID and NFC are accommodating an ultra-low energy and short-range communication topology. Hence, they are mostly applicable in the implantable devices (*Newaz et al., 2020*; *Sun, Lo & Lo, 2019*). In this way, the aggregated data at the personal server is directed to the medical server. Long-range wireless technologies such as Wi-Fi, GSM and LoRa are also used for the connection between the personal server gateway and medical sever.

The data analytics are conventionally performed at the cloud. However, cloud computing faces the issues of delay and privacy. For this reason, the term of fog computing at gateways (fog devices) was introduced in 2017 (*Rahmani et al., 2018*). This approach is used to shift some of the cloud computing works closer to the smart devices aiming at achieving a faster computation with keeping privacy (*Dang et al., 2019*; *Firouzi et al., 2018*; *Rahmani et al., 2018*). It is not deniable that using smart medical devices has made the life easier and healthier. However, there are many safety and security gaps in these devices that put not only the devices at risk but expose a significant risk to the patient's life (*Yaacoub et al., 2020*).

## Security and privacy in the IoMT

The IoMT devices are vulnerable to traditional and zero-day attacks. This is mainly due to the lack of existing security protocols and measures in the fabrication of the devices in addition to the nature of the devices and IoT network. The devices are very tiny, where their computational resources and batteries cannot handle computation of cryptographic and existing heavy security measures. Furthermore, the network of the IoMT is heterogeneous, which is composed of different protocols at each layer, making one security solution not applicable to all devices. It was reported in an estimation made by the Statista (The Statistics Portal) that the number of medical IoT devices in the European Union (EU) could reach 25.8 million devices by 2025. In addition to a steady increase in

the number of smart medical devices and benefits from the low price of wireless sensors, the security and privacy issues have become the main concern (*He et al., 2018*). Besides, when internet connected devices are increased, their produced data will eventually increase (*Dimitrov, 2016*; *Firouzi et al., 2018*; *Ma et al., 2017*). It was predicted by Statista that IoT devices could produce about 79.4 zettabytes (ZB) by 2025 (*O'Dea, 2020*). It is well known that not only the IoMT devices are at risk of cyber-attack, their data are also at a high risk. In fact, privacy issues and data disclosure are the current utmost issues in IoMT infrastructure (*Gupta et al., 2020b*; *Xu et al., 2020*).

The IoMT security and privacy requirements are different from the traditional network requirements which are usually referred to as CIA-triad (confidentiality, integrity, and availability). Other metrics such as privacy, and non-repudiation are also essential for the IoMT system (*Grammatikis, Sarigiannidis & Moscholios, 2019*; *Nanayakkara, Halgamuge & Syed, 2019*; *Newaz et al., 2020*; *Yaacoub et al., 2020*). The followings are definitions of these metrics used in the IoMT system.

a) *Confidentiality*: It limits unauthorized access to certain information and it guarantees the protection of confidential information. Unauthorized access may lead to data leakage and sometimes life-threatening situations (*Alassaf & Gutub, 2019*).

b) *Integrity*: It ensures that the data and reading of devices are not altered, deleted, or injected by unauthorized parties. Attacks on integrity such as false data injection on implantable pacemaker may lead to death (*Yaacoub et al., 2020*).

c) *Availability*: It ensures that the data, computing elements, and communications are accessible and working continuously when they are required by a service. System service interruption poses danger on patients health, considering a surgery room equipped with wireless medical devices (*Sun, Lo & Lo, 2019*).

d) *Privacy:* It ensures that privacy rules are enforced by the IoMT system and allows users to access their private details (*Mosenia & Jha, 2016*). The collection of confidential health records must conform with legal and ethical laws of privacy such as the General Data Protection Regulation (GDPR) and the Health Insurance Portability and Accountability Act (HIPAA) (*Gupta et al., 2020b*; *Tovino, 2016*). These rules ensure the protection of private patient data from disclosure. It is important that Electronic Health Records (EHRs) should not be kept beyond the time of their need. Data should not be breached while stored, transmitted, and used (*Hassija et al., 2019*). Due to open communication channels and neglecting data privacy, the possibility of confidential and private details being released, hacked, or compromised is extremely high. This can be compromised through passive attacks or active attacks (*Yaacoub et al., 2020*).

e) *Non-repudiation*: This metric encompasses the ability of the system to validate the presence or absence of an activity (*Mosenia & Jha, 2016*; *Yeh, 2016*). It ensures that the sending node is granted a delivery receipt and the receiver node is granted a proof on the identity of the sender so that none of them is denied in the process (*Newaz et al., 2020*).

**Table 2 Attacks on different IoMT layers with their respective impact on security requirements.**

| Attacks | Impact | References |
| --- | --- | --- |
| Targeted layer: Server/Database layer | | |
| Malware attack | Integrity, Availability | *Wazid et al. (2019)* |
| Ransomware attack | Integrity, Availability | *Fernandez Maimo et al. (2019)* |
| SQL injection | All | *Li et al. (2019), Gupta et al. (2020a)* |
| Social Engineering (Reverse Engineering, Shoulder-surfing) | All | *Yaacoub et al. (2020), Patel (2020), He et al. (2018)* |
| Brute Force | Confidentiality, Integrity | *Stiawan et al. (2019)* |
| Adversarial Machine Learning attacks (Causative (Poisoning and Evasion attacks), Exploratory) | Confidentiality, Integrity | *Ibitoye, Shafiq & Matrawy (2019), Mozaffari-Kermani et al. (2014), Mosenia & Jha (2016), Chakraborty et al. (2018)* |
| Targeted layer: Network layer | | |
| Denial of Service (DoS) and Distributed DoS (DDoS) | All | *Gupta et al. (2020a), Yaacoub et al. (2020), Al Shorman, Faris & Aljarah (2019)* |
| Man in the Middle (MIM) attack | Confidentiality, Integrity | *Yaacoub et al. (2020)* |
| Eavesdropping attack | Confidentiality, Non-repudiation, Privacy | *Gupta et al. (2020a), McMahon et al. (2017)* |
| Replay attack | Confidentiality, Integrity | *Spiekermann (2015), Yaacoub et al. (2020)* |
| botnet attack | Availability, Confidentiality | *Bahşi, Nõmm & La Torre (2018)* |
| Mirai attack | Availability, Confidentiality | |
| Jamming attack | Availability | *Sun, Lo & Lo (2019)* |
| Flooding attack | Availability | *Gupta et al. (2020a), Shafiq et al. (2020), Yaacoub et al. (2020)* |
| Packet Analysis attacks | Integrity, Confidentiality, Non-repudiation, Privacy | *Tseng, Wu & Lai (2019)* |
| Targeted layer: Device/sensor layer | | |
| Physical Sensor/Node tampering | All | *Xing et al. (2010)* |
| False data Injection | Integrity, Confidentiality, Non-repudiation, Privacy | *Rahman & Mohsenian-Rad (2012), Bostami, Ahmed & Choudhury (2019), Farroha (2019), Newaz et al. (2020)* |
| Resource Depletion Attacks (Battery drain, Sleep deprivation, Buffer overflow) | Availability | *Hei et al. (2010), Mosenia & Jha (2016), Gupta et al. (2020b)* |
| Side-channel | Confidentiality, Non-repudiation, Privacy | *Zhang, Raghunathan & Jha (2013), Newaz et al. (2020)* |
| Hardware Trojan | All | *Qu & Yuan (2014), Mosenia & Jha (2016)* |
| Eavesdropping | Confidentiality, Non-repudiation, Privacy | *Gupta et al. (2020a), McMahon et al. (2017)* |

**Note:**
Each row represents a different type of attack, and the rows show their targeted layer, impact, and reference.

A threat that risks any of the aforementioned pillars of the IoMT security and privacy is considered an attack. Table 2 shows the categorized attacks based on the IoMT targeted layer with respect to their impacts on the security requirements of the IoMT system. It is worth to mention that despite of the reported attacks the appearance of zero-day attacks is a daily possibility. While these attacks are threatening the privacy of patients, it is also resulted in unbearable financial damage and loss of reputation (*Sun, Lo & Lo, 2019*). For example, a recent ransomware attack on hospitals in the USA, which infected 400 U.S. sites has resulted in service disruption and data disclosure (*Davis, 2020b*).

According to a latest analysis by Comparitech, these attacks have cost the healthcare industry over $160 million since 2016 (*Davis, 2020a*). Additionally, it was claimed that attacks on brain implants, lead to death (*Rathore et al., 2019*). Therefore, with the increased risk of cyber-attack on the IoMT system, the emergence and development of robust security solutions have become a must.

## Machine learning techniques and their evaluation metrics

Machine Learning (ML) is a branch of Artificial Intelligence (AI) which learns from data and experiences without being explicitly programed (*Kubat, 2017*). ML can play a vital role in the IoMT, especially at computing nodes such as cloud/fog computing, for continuous data analysis and producing meaningful information. Recently, ML has been interestingly applied in various areas of the IoT and IoMT (*Alimadadi et al., 2020*; *Ardabili et al., 2020*; *Cui et al., 2018*; *Durga, Nag & Daniel, 2019*; *Pannu, 2015*; *Pramanik et al., 2017*; *Xiao et al., 2018*). ML can be effectively used in many ways to tackle the security issues of the IoMT (*Cui et al., 2018*; *Gupta et al., 2020a*; *Pandey et al., 2020*; *Pirbhulal et al., 2019*), whereas traditional security measures cannot prevent the system from zero-day attacks, and they are resource extensive for the IoMT systems. Advanced ML techniques can learn from massive generated data of the IoMT, thereby finding new trends of attacks (*Hussain et al., 2020*). Hence, data hungry ML techniques can be perfectly tuned with the IoMT.

There are three main types of ML algorithms which can be used for solving the IoMT security problems, namely **supervised, unsupervised,** and **semi-supervised** algorithms. The ***Supervised*** technique deals with labeled data, in which classes or labels are known. There are two forms of supervised learning which are classification and regression (*Kubat, 2017*). Examples of supervised learning are Support Vector Machine, Decision Trees and Neural Network. These methods are used for the **attack and malware detection** purpose such as signature based Intrusion Detection System (IDS) (*Eskandari et al., 2020*; *Swarna Priya et al., 2020*). Since it is not always easy to label the data by human, ***unsupervised techniques*** are utilized to categorize them based on their similar traits. Examples of these techniques are K-nearest Neighbor (KNN), Self-Organizing Map (SOM), and Latent Dirichlet Allocation (LDA) (*Gupta et al., 2020a*). These methods are effective for **anomaly detection** of new attacks (*Ahmad et al., 2019*; *Bostani & Sheikhan, 2017*; *Eskandari et al., 2020*). Based on the "no free lunch" theory, not every unlabeled data is useful for prediction (*Rathore & Park, 2018*). Hence, one can overcome this issue by incorporating some labeled data. ***Semi-Supervised Learning*** is a younger ML sub-field, in which some data are labeled out of a wide number of learning data (*Wolfgang, 2011*). Also, semi-supervised models can be used for attack detection and to avoid adversarial attacks on ML techniques (*Miyato et al., 2018*; *Rathore & Park, 2018*). In addition to the main classes of ML, a new branch called ***Deep learning (DL)*** has emerged recently, which is an advanced form of Neural Network. It has several layers of artificial neural network that mimic the working principle of human brain in processing data for detecting objects, recognizing speech, translating languages, and making decisions. DL can

perform feature selection/extraction based on its learning process without the need of another method compared to traditional ML techniques that require an additional feature selection (*Amanullah et al., 2020*). DL comes in different forms, including Convolution Neural Network (CNN), Recurrent Neural Network (RNN), Long-Short Term Memory (LSTM), Auto-Encoders (AE), Boltzmann Machine (BM), Generative Adversarial Networks (GAN), Feed forward Deep Networks (FDN) and Deep Belief Networks (DBN) (*Amanullah et al., 2020*; *Hussain et al., 2020*; *Saeed et al., 2016*). Auto-encoders are usually the best choice of unsupervised learning to be used for anomaly detection (*Lopez-Martin et al., 2017*). The use of DL for security solutions such as attack detection could be a durable method for slight mutations or new threats due to its ability to extract features (*Amanullah et al., 2020*; *Diro & Chilamkurti, 2018*; *Durga, Nag & Daniel, 2019*; *Ibitoye, Shafiq & Matrawy, 2019*; *Parra et al., 2020*; *Sollins, 2018*). On the other hand, the IoMT sensors generate a large amount of data at a fast speed, thereby producing Big data (*Cao, 2017*; *Pramanik et al., 2017*; *Saheb & Izadi, 2019*; *Marjani et al., 2017*). **Big data** refers to the data that are too large and/or complex to be efficiently handled by conventional technologies and tools (*Dimitrov, 2016*; *Hameed, Hassan & Muhammad, 2017*; *Sadoughi, Behmanesh & Sayfouri, 2020*; *Vinitha et al., 2018*). A big data is usually defined by three key features, known as 3Vs, which are volume, velocity and variety (*Amanullah et al., 2020*; *Russom, 2011*). It was seen in literature that DL and big data have been used together as DL performs better with increasing the amount of data (*Gheisari, Wang & Bhuiyan, 2017*). Very recently, DL and Big Data were used in combination with new technologies of blockchain and reinforcement learning for security purposes in the industrial IoT systems (*Aman et al., 2020*; *Liang et al., 2020*; *Liu, Lin & Wen, 2018*). Concludingly, the ML methods can be employed in different security solutions such as:

### Sensor anomaly detection in medical devices

Anomaly detection is the process of finding deviations from the normal data. Such anomalies are attributed to the occurrence of phenomena that do not obey a regular process. These irregularities are triggered by abnormal behaviors such as internal assault or false data injection attack (*Amanullah et al., 2020*), leading to a false reading of the sensors. ML approach in anomaly detection is the most used mechanism for sensor security. Nonetheless, it needs a considerable adoption in order to be implemented in the IoMT (*Butun, Kantarci & Erol-Kantarci, 2015*; *Hasan et al., 2019*).

### Authentication and access control

Authentication is the process of ensuring whether the IoMT user is correct, while authorization is associated with the extent of access given to each individual (patient and healthcare provider) (*Aghili et al., 2019b*). They are the most effective methods to prevent attacks on confidentiality and integrity of the IoMT data (*Aghili et al., 2019b*; *Nguyen et al., 2019*; *Yang et al., 2018*). Different techniques can be combined in the process of authentication and authorization such as cryptography, ML, and lightweight approaches (*Aghili et al., 2019b*; *Shen et al., 2018*; *Wu et al., 2018*). Details about these methods can be found in *Sun, Lo & Lo (2019)*.

**Table 3 The evaluation metrics used for ML techniques.**

| Metric | Description | Formula |
|---|---|---|
| Accuracy | Determines the performance of the model in recognizing all classes, respectively | $Acc = \dfrac{TP + TN}{N}$ |
| Sensitivity (Recall) | Measures the completeness, which is the percentage of positive predicted samples to the positive samples in dataset is depicted. | $Sensitivity = \dfrac{TP}{TP + FN}$ |
| Specificity (Precision) | Shows the exactness, in which the percentage of correctly positive predictive samples to all positive predictive samples by the model are calculated. | $Specificity = \dfrac{TP}{TP + FP}$ |
| False Positive Rate (FPR) | Measures the number of those normal network behaviors which are calculated as attack. | $FPR = \dfrac{FP}{FP + TN}$ |
| Performance overhead | It is the calculation of any combination of (memory, CPU, energy) overhead taken by the ML techniques to perform a task. | *Big(o)notation for time and space complexity and energy unit for energy usage* |
| *FN* | | |

**Notes:**
Each row represents a metric, and the columns show their description, and formula.
TP, True Positive; TN, True Negative; N, total number of samples; FN, False Negative; FP, False Positive.

### Intrusion and malware detection

The most common intrusion detection is network intrusion detection (NIDS) using network traffic analysis. If the log data of sensors and devices are used, it becomes host-based intrusion detection (HIDS) (*Aluvalu, 2020*). Both types of intrusion detection use metrics for finding anomalies (unknown-attacks) and signatures (known-attacks) that do not match with the normal network traffic or sensor data. These detection systems are similar in having an agent for collecting data and a processing unit in the attack detection and reporting the intrusions. However, they are different in: (i) Source of data: host-based (*Asfaw et al., 2010*), network-based (*Maleh et al., 2015*), hybrid (*Ahmad et al., 2019*), (ii) Method of detection (*Abhishek et al., 2018*): signature-based (*Wang et al., 2018*), anomaly-based, and (iii) Architecture: centralized, and distributed (*Anthi, Williams & Burnap, 2018*; *Da Costa et al., 2019*). Signature based approaches can easily detect the known attacks but unable to detect new attacks. On the other hand, anomaly-based IDS find anomalous behavior by learning from current data. Therefore, they can detect new attacks, but they are less accurate and computationally expensive (*Arshad et al., 2020*; *Jan et al., 2019*; *Midi et al., 2017*). The recent attacks such as DoS, DDoS and different malware attacks are the most common mitigated attacks by IDSs (*Din et al., 2019*; *Meidan et al., 2018*; *Roopak, Tian & Chambers, 2020*).

In the process of evaluating ML techniques in terms of accuracy, reliability, and applicability, some special metrics should be considered. Table 3 shows the most used metrics with their definition and formula.

## IoMT security and privacy solutions using machine learning
### Research rationale

The unique nature of the IoMT system with small size devices, heterogeneous network, and diverse protocols, has made the implementation of traditional security frameworks difficult for the medical companies. This in turn makes the IoMT system susceptible to different attacks. Recent advancement in the techniques and technologies of ML has led to

achieve fruitful strategies to tackle the issues of the IoMT security. However, the IoMT system is facing more challenges than ordinary network systems.

Internet of Medical Thing devices usually generate a large amount of streaming data (*Amanullah et al., 2020*). The diversity nature of these data along with the limited power and resources of the IoMT devices, especially for the implantable medical devices (*Sun, Lo & Lo, 2019*), impose a high computational burden on the traditional ML techniques, thereby reducing their effective application in the IoMT devices. Hence, new strategies are required to apply the ML approaches efficiently.

Consequently, understanding the current security and privacy issues of the IoMT system with their respective solutions using ML techniques is essential. Furthermore, it is significant to know the effectiveness of current deployed ML techniques and to understand their solution strength offered to the challenges of the IoMT system so far. We found that little attention has been paid in literature to elaborate on these issues. Therefore, in this work, a Systematic Literature Review (SLR) is presented, in which attempts are made to reveal the strong points and limitations of the works performed in this area followed by establishing robust improvement strategies. As such, three research questions can be generated as follows:

RQ1: What is the current state of the art and direction of study in the IoMT security using ML?
RQ2: What kind of tools and data are used for applying ML techniques in the IoMT security?
RQ3: How ML techniques are effectively applied and what are their limitations?

This study is intended for new researchers in the field, and for those who are keen to know about recent advances and limitations in the IoMT security issues followed by their solutions using ML approaches.

### Related studies

In a review performed by *Cui et al. (2018)*, an overview of ML application in the domain of generic IoT was reported to focus on the main applications of ML and its relevant techniques in IoT. However, the work has partially covered the IoT security solutions with ML. In another study, *Tahsien, Karimipour & Spachos (2020)* reviewed the architectures of generic IoT and ML-based potential solutions for the IoT security. Another comprehensive review on using ML techniques in generic IoT security was also conducted by *Hussain et al. (2020)*. The authors discussed major threats to each layer of the IoT and reviewed recent works that have used ML techniques for securing IoT. Noticeably, the reviewed studies, tools and datasets were not comprehensively elaborated and the IoMT system was not explored. A review study on security and privacy of the IoMT was conducted by *Sun, Lo & Lo (2019)* revealing the security requirements and challenges of the IoMT with more focus on authentication and access control (*Sun, Lo & Lo, 2019*). Moreover, a survey on the IoMT security and privacy was carried out by *Newaz et al. (2020)*, in which a detailed discussion was given on the security and privacy threats in healthcare systems. They also presented a subsection on the current solutions for healthcare IoT security. On the other hand, a review on the IoMT security issues and limitations with details about the attacks and their impact on the IoMT was presented,

**Table 4 Comparison between this survey and other related surveys.**

| Year | References | IoT domain | Architecture | Threats | ML methods | Big data | ML for IoMT security | Systematic analysis |
|------|-----------|-----------|-------------|---------|-----------|----------|---------------------|--------------------|
| 2018 | Cui et al. (2018) | Generic | NA | NA | Discussed | NA | NA | NA |
| 2020 | Tahsien, Karimipour & Spachos (2020) | Generic | IoT architecture | IoT attacks | Discussed | NA | NA | NA |
| 2020 | Hussain et al. (2020) | Generic | NA | IoT attacks | Discussed | Big data at cloud | NA | NA |
| 2019 | Sun, Lo & Lo (2019) | IoMT | IoMT architecture | IoMT Security requirement | NA | NA | Partially discussed | NA |
| 2020 | Newaz et al. (2020) | IoMT | IoMT architecture | IoMT attacks | NA | NA | Partially discussed | NA |
| 2020 | Yaacoub et al. (2020) | IoMT | IoMT architecture | IoT attacks | NA | NA | NA | NA |
| – | This study | IoMT | IoMT architecture | IoMT attacks | Discussed | Discussed | Discussed | Comprehensive and Systematic Review |

**Note:**
Each row shows a different related study, and the columns show their features.

whereas a special attention was paid on lightweight security solutions (Yaacoub et al., 2020). Concludingly, there have been some reviews found in literature about generic IoT security using ML and DL. However, little attention has been paid to ML applications for the IoMT security and privacy. To this end, the current review is intended to address the role of ML technologies in tackling the issues of the IoMT security and privacy. That is by carrying out a comprehensive and systematic review on the related works performed in literature. For this reason, it is difficult to compare our work with the existing surveys. However, a fair relevant comparison of the previous reviews with that of the current study is given in Table 4.

# SURVEY METHODOLOGY

## Research design
In the current work, a systematic research design is generated. After creating a list of research questions, searching for relevant papers was started from different databases including IEEE, Web of Science, Springer Link, Scopus, Science Direct. Then, the most specific and relevant papers were extracted to answer the research questions. Later on, the selected papers were comprehensively screened and analyzed. Finally, the results were presented using different methods.

## Database searching and research selection
A keyword-based search was applied by using different forms and combinations of Machine Learning, The Internet of Medical Things, Security and Privacy and their synonyms. Then, the synonym list was expanded during researching and a Research Information Template (RIT) was generated accordingly, as shown in Fig. 1. Boolean expressions with all the keyword combinations were constructed to form all the searching possibilities. The literature searches were carried out in April 2020, whereas each RIT query string was

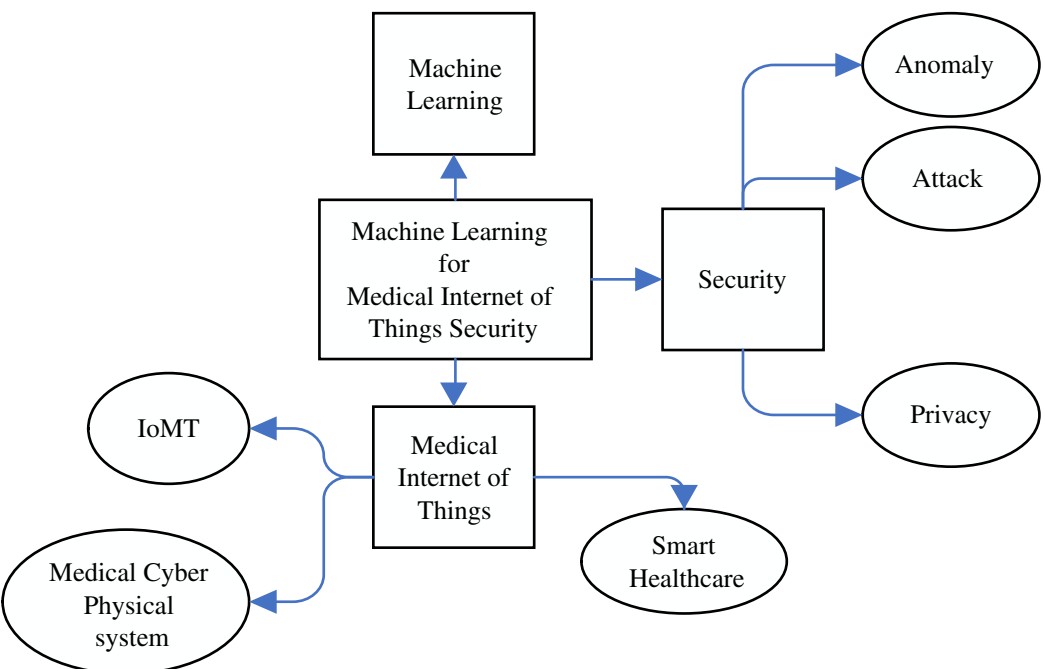

**Figure 1 The keywords used in the Research Information Template (RIT).** The entire Mind Map shows the keywords used in the Research Information Template (RIT). The rectangular box at the middle represents the main keywords, while the square boxes represent the derived similar words from the main words. The oval shaped keywords are dervied from their previous sequare box keywords.

checked in the respective databases followed by saving the query results. The period affecting this analysis was justified between January 2010 and April 2020. Initially, the full metadata would have been selected for searching and if this option was not available, the common search choice (keyword, title and abstract) was used. The search queries and results returned by the databases are shown in Table 5. Noteworthy, a total of 28,155 search results were returned from all the five databases. The titles were then examined for significant and duplicate papers and a total of 500 papers were remained. After that, the papers whose mentioned IoMT/ML/security and privacy were chosen. Among these, 180 papers underwent a deeper scanning on their abstract and conclusion, considering the paper quality, whether they are from refereed journals or other resources. As such, a total of 153 publications were filtered and the remainder was removed. After going through the full text of the papers, only 43 papers were selected for critical analysis and conducting a careful review of their contents. The rest of the papers were used while discussing the topics related to the background of the study. The selection criteria given in Table 6 were performed equally on the title, abstract, conclusion, and full paper in all stages of screening, while the detailed process of the methodology is shown in Fig. 2.

## RESULTS AND FINDINGS

In this work, two types of analysis were performed which include a bibliometric analysis and technical analysis, as discussed in the next subsections.

**Table 5 The searching queries and results achieved from five different databases.**

| Searching texts vs databases | IEEE Xplore | SpringerLink | Scopus | Science direct | Web of science |
|---|---|---|---|---|---|
| Machine Learning AND Medical Internet of Things AND Security | 89 | 3,431 | 63 | 2,459 | 33 |
| Machine Learning AND Medical Internet of Things AND Privacy | 37 | 1,093 | 43 | 1,494 | 31 |
| Machine Learning AND Medical Internet of Things AND Intrusion | 6 | 336 | 2 | 445 | 3 |
| Machine Learning AND Medical Internet of Things AND Attack | 13 | 1,225 | 2 | 1,203 | 18 |
| Machine Learning AND IoMT AND Security | 9 | 81 | 3 | 929 | 5 |
| Machine Learning AND IoMT AND Privacy | 8 | 44 | 3 | 78 | 6 |
| Machine Learning AND IoMT AND Intrusion | 0 | 8 | 1 | 21 | 1 |
| Machine Learning AND IoMT AND Attack | 0 | 33 | | 64 | 5 |
| Machine Learning AND Medical Cyber Physical system AND Security | 30 | 481 | 5 | 1,038 | 11 |
| Machine Learning AND Medical Cyber Physical system AND Privacy | 8 | 429 | 3 | 693 | 4 |
| Machine Learning AND Medical Cyber Physical system AND Intrusion | 6 | 321 | 3 | 301 | 4 |
| Machine Learning AND Medical Cyber Physical system AND Attack | 9 | 494 | 0 | 699 | 11 |
| Machine Learning AND Smart healthcare AND Security | 57 | 1,971 | 12 | 2,134 | 52 |
| Machine Learning AND Smart healthcare AND Privacy | 25 | 1,012 | 3 | 1,901 | 36 |
| Machine Learning AND Smart healthcare AND Intrusion | 7 | 334 | 1 | 431 | 21 |
| Machine Learning AND Smart healthcare AND Attack | 20 | 1,031 | 3 | 1,209 | 20 |
| Total including duplicates | 324 | 12,324 | 147 | 15,099 | 261 |

Note:
Each row shows different queries used for all databases, and the columns show their results. The searching queries and results achieved from five different databases.

**Table 6 Selection criteria of the papers at final stage.**

| Criteria# | Questions | Answer |
|---|---|---|
| 1 | Does the paper relevant to the topic? | Y/N |
| 2 | Does the work propose a machine learning related solution and method to solve a problem in the IoMT security and/or privacy? | Y/N |
| 3 | Is the paper published in scholarly journals, conferences, books? | Y/N |

Note:
Each row represent a criteria for selecting the papers, and the column shows the response.

## Bibliometric analysis

To show the leading countries whose researchers working in the field of the IoMT and its security, each individual paper was examined according to the affiliation of authors. It was observed that the USA has made 30% of the papers among 21 affiliated countries, as shown in Fig. 3. Moreover, it was found that research performed on ML applications in the IoMT security has grown steadily from 10 percent in 2015 to a peak value of 80 percent in 2019, as shown in Fig. 4.

It is worth to mention that the majority of the papers were journal articles, contributing to 73% of the papers, while 21% of the papers were from conferences of the Institute of Electrical Electronics Engineering (IEEE) and only 6% of the papers belonged to book chapters as seen in Fig. 5. Furthermore, a significant number of papers (64%) was published by the IEEE and 17% was published by Elsevier, as seen in Fig. 6.

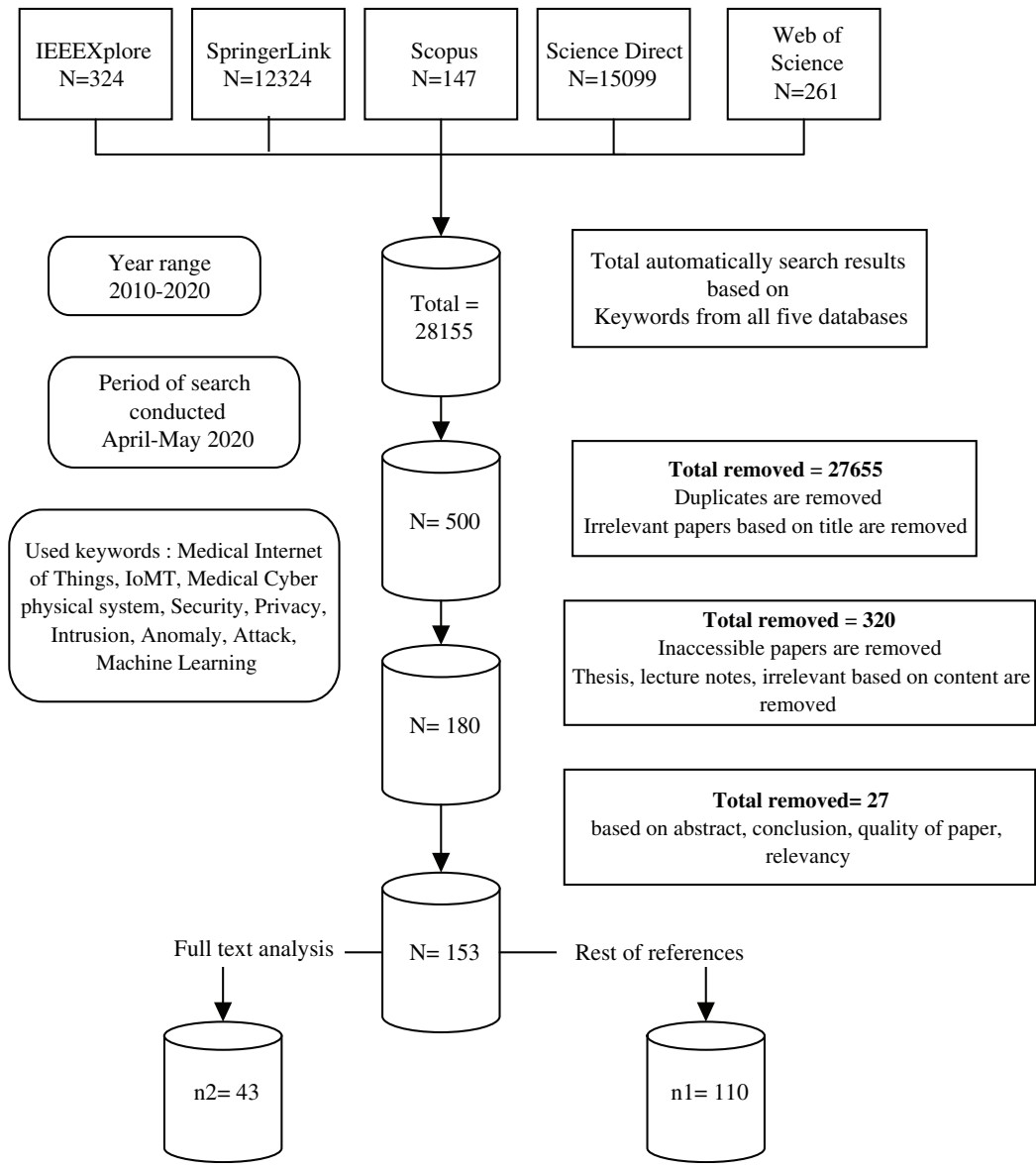

**Figure 2 The search strategy used for selecting the research papers based on the PRISMA guideline.**
The flow chart represents the procedure of searching in different databases using PRISMA guideline.
It starts from top to bottom, showing each step of the paper selection and fileteration.

## Studies technical analysis

### Classification of studies

The vast majority of ML related articles were about supervised ML, while the rest of papers were found to report on a combination of supervised and unsupervised ML, with few papers to focus on unsupervised ML. Deep learning was also used in some of the studies, while one study has used big data technology. The papers were further categorized based on the type of medical devices intended to get secured. Almost all studies have focused on the security of wearable devices, while few of them elaborated on the security

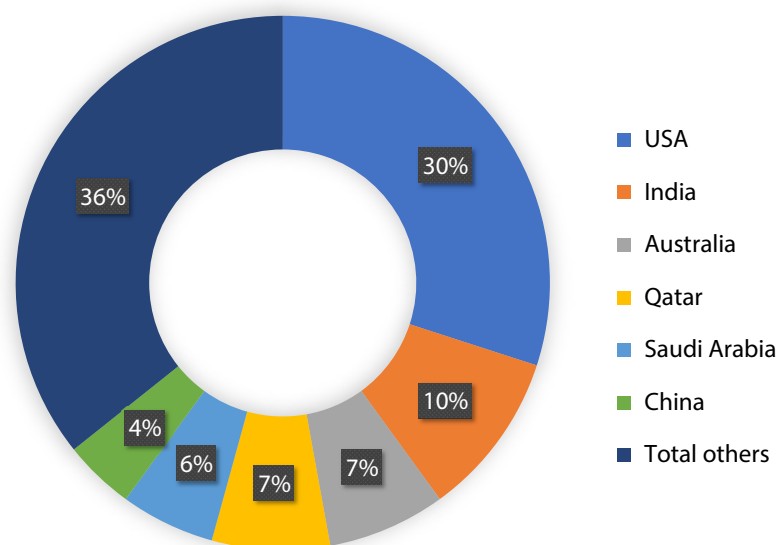

**Figure 3 Geographical distribution of the papers.** The pie chart shows the percentage of the papers by each country.                                               

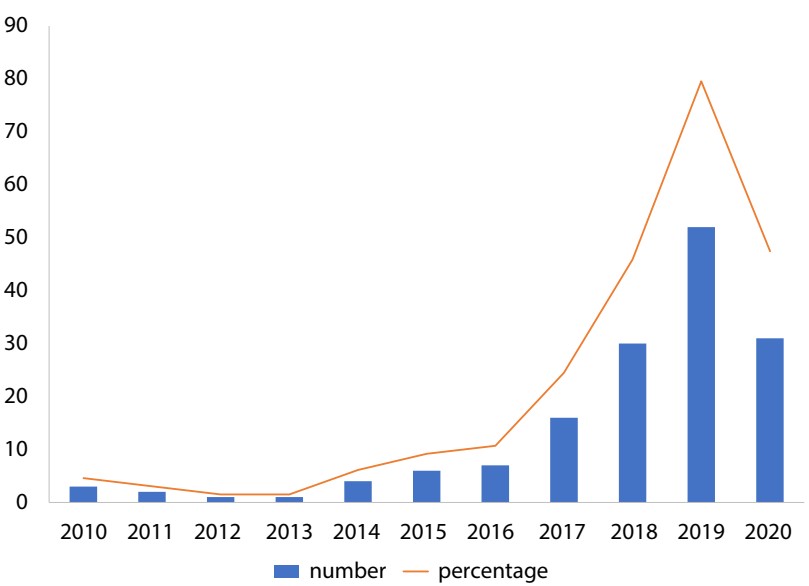

**Figure 4 Distribution of the papers by year.** Each Blue Bar represents the number of papers published in each year and the orange line shows the percentage of the reviewed papers in each single year.

measures for implantable devices. Only one study was found to focus on securing programmer devices. The targeted IoMT layers in most of the studies were device and sensor layers. Network layer was addressed in some studies, while cloud layer was reported by two paper. Table 7 shows further details on the disseminated studies related to their ML category and types of devices with their targeted layer.

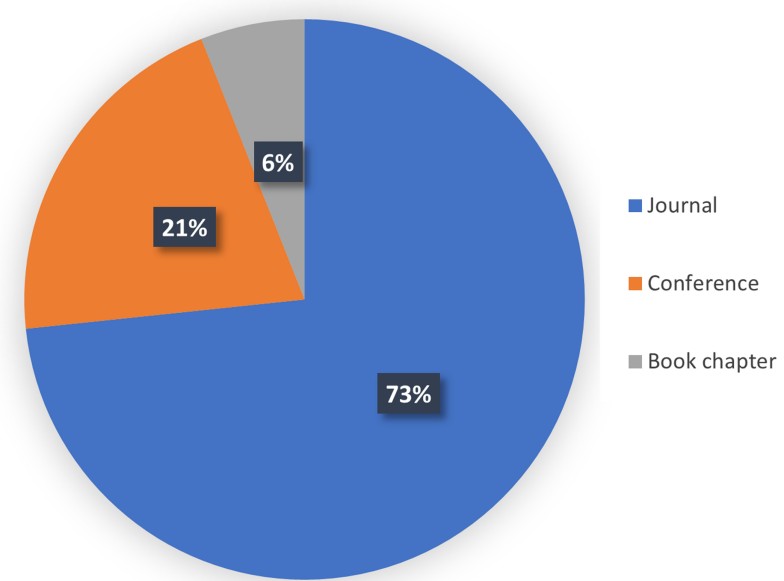

**Figure 5 The type of analyzed papers used in the current research.** The pie chart shows the percentage of the analyzed papers in each catgory of Journal, Conference, and Book chapter.

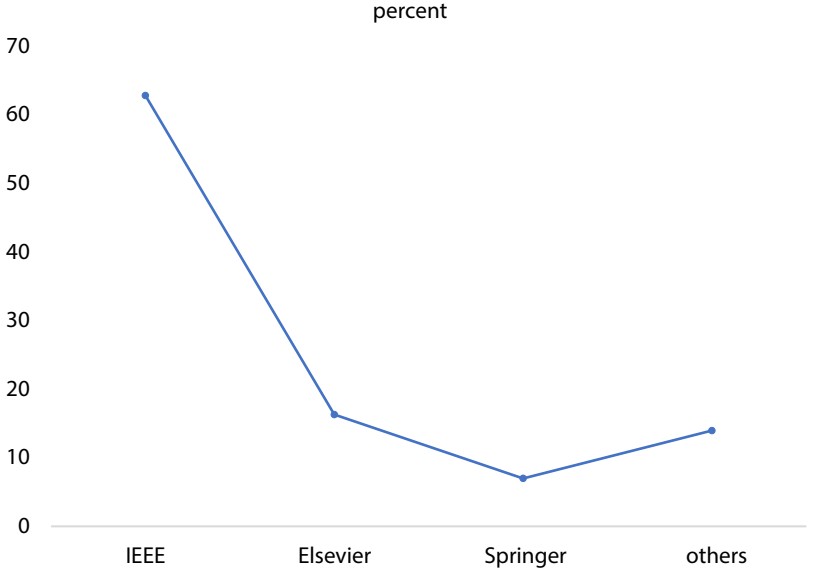

**Figure 6 Distribution of the papers according to the publishers.** The curve shows the percentage of the analyzed papers that were published by each publisher.

The papers were then classified into different subsections based on the approaches taken to tackle the security issues of the IoMT, as was discussed in the background section. In addition, an in-depth assessment was carried out through a critical analysis of the articles, demonstrating the strong characteristics and limitations of each study. In the following sections, we discuss on the findings of the aforementioned analysis.

**Table 7 Classification of papers based on ML category, medical device, and the IoMT layer.** The table shows a matrix representation of the paper's classification to different categories. Classification of papers based on ML category, medical device, and the IoMT layer.

| ML category—medical device category | IoMT layer-References |
| --- | --- |
| Supervised ML—Implantable, wearable | **Sensor layer**-{*Haque, Rahman & Aziz (2015)*, *Abdaoui et al. (2020)*, *Rathore et al. (2018c)*, *Khan et al. (2017)*, *Gao & Thamilarasu (2017)*, *Newaz et al. (2019)*, *Ben Amor, Lahyani & Jmaiel (2020)*, *Mohamed, Meddeb-Makhlouf & Fakhfakh (2019)*, *Salem et al. (2014)*, *Hau & Lupu (2019)*, *Nagdeo & Mahapatro (2019)*, *Verner & Butvinik (2017)*}<br>**All layers**-*Rathore et al. (2018b)* |
| Supervised ML—Wearable, smart watches, smart fitness | **Device layer**-{*Pirbhulal et al. (2019)*, *Mawgoud, Karadawy & Tawfik (2019)*, *Barros et al. (2019)*, *Zhang et al. (2018)*, *Shang & Wu (2019)*, *Musale et al. (2019)*, *Vhaduri & Poellabauer (2019)*, *Mohsen, Ying & Nayak (2019)*, *Rathore et al. (2018a)*}<br>**Network layer**-{*Begli, Derakhshan & Karimipour (2019)*, *Itten & Vadakkumcheril (2016)*, *Schneble & Thamilarasu (2019a)*, *Odesile & Thamilarasu (2017)*, *Swarna Priya et al. (2020)*, *Asfaw et al. (2010)*, *Alrashdi et al. (2019)*, *Wazid et al. (2019)*, *Fernandez Maimo et al. (2019)*}<br>**Cloud layer**-{*Punithavathi et al. (2019)*, *Landau et al. (2020)*} |
| Unsupervised ML—Implantable, wearable | **Sensor layer**-{*Ben Amor, Lahyani & Jmaiel (2020)*, *Sehatbakhsh et al. (2018)*, *Zhang, Raghunathan & Jha (2013)*, *Verner & Butvinik (2017)*} |
| Unsupervised ML—Wearable | **Device layer**-{*Zhang et al. (2018)*, *Shang & Wu (2019)*, *Musale et al. (2019)*, *Vhaduri & Poellabauer (2019)*, *Mohsin et al. (2019)*}<br>**Network layer**-{*Schneble & Thamilarasu (2019b)*, *He et al. (2019)*, *Swarna Priya et al. (2020)*, *Thamilarasu (2016)*, *Fernandez Maimo et al. (2019)*} |
| Unsupervised ML—ICD, Programmer | **Device layer**-*Kintzlinger et al. (2020)* |
| Deep learning—Wearable | **Device layer**-{*Swarna Priya et al. (2020)*, *Manimurugan et al. (2020)*, *Shakeel et al. (2018)*, *Rathore et al. (2019)*, *Mohsin et al. (2019)*} |
| Big data—Wearable | **Device layer**-*Zhang et al. (2018)* |

## Sensor anomaly detection for medical devices

Herein, the review results of the current subsection related research articles are presented and analyzed, while the highlighted limitations of the studies are given in Table 8.

In a work performed by *Haque, Rahman & Aziz (2015)*, a sensor anomaly detection system was proposed to differentiate true from false alarms. The research used a historic data to be compared with the actual sensed data for prediction, whereas majority voting was used for their distinguishing. Consequently, the error was calculated based on dynamic threshold. The proposed method has been implemented in Java environment, supplied by the SMO regression. The results illustrated that the proposed system had a high Detection Rate and low FPR for three medical datasets. Furthermore, referring to the security of signals from deep brain stimulators, the authors in *Abdaoui et al. (2020)* built a system for distinguishing false alarms from legitimate ones and classified the attacks using Raspberry Pi3 and deep learning. It was found that deep learning can show an accuracy of about 97% to learn and predict the fake signals. Also, a web-based application was generated using the web engine (Flask) for that purpose.

Despite the effective application of ML algorithms, they are generating high computational overhead on the low-power embedded frameworks. *Rathore et al. (2018c)* presented a neural network based MLP solution embedded on an FPGA chip system for securing insulin pump devices that are used by diabetic patients. The authors reported an accuracy of 98.1% for their system in distinguishing fake from genuine glucose

**Table 8 Details of published studies on anomaly and attack detection to the sensors/medical devices.**

| Ref. | Methods | Detection type | Good features | Limitations | Tools | Dataset info. |
|---|---|---|---|---|---|---|
| *Haque, Rahman & Aziz (2015)* | SMO | Anomaly detection | -high detection rate<br>-low FPR | -high computation overhead | Weka | -10 real datasets<br>-(MIMIC)data |
| *Abdaoui et al. (2020)* | Deep Learning | Anomaly based false alarm detection | -real time<br>-high accuracy | -high computation overhead<br>-high FPR | Tensor flow and Keras in Python | John Radcliffe Hospital data |
| *Rathore et al. (2018c)* | neural network-based MLP | Anomaly based false alarm detection | -real time<br>-energy efficient<br>-high accuracy<br>-reliable | -high memory requirement due to training overhead | -NI myRIO<br>-FPGA-based MLP | UCI Diabetic dataset |
| *Khan et al. (2017)* | DWT and Marcov model | Anomaly based false data detection | -high detection rate<br>-high TNR<br>-real time | -detection rate decreases when there is too much attack<br>-high FNR and FPR | MATLAB | The ECG dataset from MIT-PHYSIOBANK |
| *Gao & Thamilarasu (2017)* | Decision tree, SVM and K-means | Anomaly-based attack detection | -high accuracy<br>-low FPR<br>-low training time<br>-low prediction time | -no validation<br>-memory, battery usage is not considered<br>-fails in detecting insider attack | Castalia | Simulation data |
| *Sehatbakhsh et al. (2018)* | (K-S test) on external hardware device | malware Anomaly detection | -high TPR<br>-low FPR<br>-low detection latency<br>-no overhead on the medical device | -external device needs maintenance and the device itself could be hacked (stolen or lost) | -Open Syringe Pump<br>-Arduino UNO, Nios-II, OlimexA13, and TS-7250 | Testbed data |
| *Zhang, Raghunathan & Jha (2013)* | a model is embedded on an external device | multi-layered anomaly detection. | -zero overhead on battery<br>-real time<br>-multiple attacks detection<br>-hybrid detection | -protecting only integrity<br>-external device needs maintenance and it could be hacked (stolen or lost)<br>-lack of behavioral anomaly detection experiment | USRP Glucose monitoring and insulin delivery systems | Testbed data |
| *Newaz et al. (2019)* | ANN, DT, RF, and k-NN | Anomaly detection using medical device data | -high accuracy<br>-high F1<br>-No overhead on the sensors<br>-uses body functioning data | -high training overhead<br>-performance overhead not calculated | MATLAB | A set of heath dataset from different sources |
| *Ben Amor, Lahyani & Jmaiel (2020)* | PCA and Correlation Coefficient | Anomaly based faulty sensor data detection. | -real time<br>-lightweight<br>-improved accuracy<br>-improved FPR | -energy, CPU usage is not considered<br>-lacks attack detection at server and transmission | -AUDITmodule<br>-Java and R languages | MIMIC database |
| *Mohamed, Meddeb-Makhlouf & Fakhfakh (2019)* | Statistical signal amplitude calculation | Anomaly based intrusion cancelation | -using more than one type of Sensor type<br>-high TPR | -not lightweight<br>-not real time<br>-high performance overhead | MATLAB | real medical ECG and EMG datasets |

*(Continued)*

| Ref. | Methods | Detection type | Good features | Limitations | Tools | Dataset info. |
|---|---|---|---|---|---|---|
| *Salem et al. (2014)* | Classifiers (SVM, RF, K-NN, Decision Trees) and Regression | Anomaly detection for sensor physiological data. | -Low detection time -High TPR -Low FNR | -Not lightweight -Performance overhead not calculated | Weka | MIMIC dataset |
| *Hau & Lupu (2019)* | Time series approach temporal-attribute correlations | Anomaly based false data injection attack detection. | -High detection for single sensor -Acceptable detection for moderate sensors | -Long detection time -Disable when many sensors increase (collusion) | Not given | MIMIC dataset |
| *Nagdeo & Mahapatro (2019)* | combined ANN with Ensemble LinReg | Classification of anomalous and faulty sensor physiological data | -High TPR -Low FPR | -Not real-time -Not lightweight -Unknown dataset source -Performance overhead not calculated | Weka | ECG dataset |
| *Verner & Butvinik (2017)* | Otsu's Thresholding, and Linear SVM | Sensor data modification detection. | -High precision -High recall -Can be used for other IoT domains | -Not real-time -Not lightweight -Performance overhead not calculated | MATLAB | (JDRF) (CGM) Clinical Trial dataset |
| *Kintzlinger et al. (2020)* | statistical and One class SVM | Rule and knowledge based anomaly attack detection. | -High TPR -Low FPR -High accuracy -Real time | -Layer III-V is useless -Heavy training overhead -Performance overhead not calculated -Learns from only benign data | Not given | Self-created clinical data |
| *Rathore et al. (2019)* | Deep Learning classifier | Attack prediction | -Classify different attacks -Reduced training time | -Not lightweight -Accuracy not given | Keras with Theano in Python | Parkinson tremor dataset from Physionet |

**Note:**
Each row represent each paper under category anomaly and attack detection to the sensors/medical devices, and the columns show the characteristics that are used for evaluating them.

measurements. The reliability of whole framework was improved by 18% in the case of securing one device and enhanced by 90% in the case of securing the whole devices.

*Khan et al. (2017)* proposed a personal server centered (phone based) Markov model-based detection mechanism for the detection of multiple intrusions such as forgery attacks, false data insertions, and data modifications in the ECG data for smart medical devices. The extracted features by DWT method were generated representing a feature set followed by their division into sequences. Each sequence's probability was then calculated.

The probability value was used as a benchmark to decide if any changes have occurred. Analyzed results showed that the method has a high detection rate with abnormalities of 5% and 10% in the dataset. However, it has a higher TNR with reduced running time for both 5% and 10% abnormalities.

In another study (*Gao & Thamilarasu, 2017*), some ML techniques were used, including decision tree, SVM and K-means, to detect the security attacks in implantable devices.

An external detection device was used to monitor the network and the ML classifiers were utilized to detect anomalies on the gateway device for detecting forced device authentication that results in resource depletion of the device. For this purpose, a feature set specific to IMD devices was constructed. Experimental results demonstrated that decision tree-based algorithms achieved the highest detection accuracy, low false positive rate, fast training and prediction speed compared to those of other algorithms.

In another study made by *Sehatbakhsh et al. (2018)*, SYNDROME was proposed. This method can detect code injection attack in a known program which runs on the system in a real time manner. Statistical based methods such as K–S test and external hardware device were used for detecting signal anomaly. The ability of the method was evaluated by implementing control-flow hijack attacks on a real medical device (syringe pump) embedded system. The evaluation results on using four distinct hardware systems have shown that the proposed model can detect all the attacks with 100% TPR and zero false positive, while the detection latency was less than 2 ms.

In a pioneer work carried out by *Zhang, Raghunathan & Jha (2013)*, a security framework was proposed for medical devices monitoring (MedMon). The proposed model was embedded on an external device which listens to all the passed signals coming from or sending to the medical devices by using a multi-layered anomaly detection (behavioral and physical anomalies). The system is useful for those medical devices that do not use encryption. Consequently, the framework either passively notifies the user or actively jams the signal. This solution does not add power overhead on the medical devices without modification to their software and hardware. An insulin delivery device was tested against the proposed method. Results depicted that the system could successfully detect multiple attacks.

For the same purpose, a different approach has been proposed by *Newaz et al. (2019)* which is based on ML data-driven security framework, called HealthGuard, for detecting three types of malicious activities in a SHS by considering interconnected body function. Here, ML based techniques (Artificial Neural Network, Decision Tree, Random Forest, and k-Nearest Neighbor) were used to interpret the physiological signs in multiple attached SHS instruments and compare them to identify the differences in the person's body functions, thereby differentiating benevolent and malicious behaviors. Moreover, there is no need to have user identification for the medical devices and the framework does not increase any overhead on the sensors while collecting data. The proposed system is trained with physiological data obtained from eight IoMT devices containing 12 genuine events consist of 7 normal patient activities and five disease associated activities. Results showed an accuracy of 91% and F1 score of 90%.

*Ben Amor, Lahyani & Jmaiel (2020)* suggested an anomaly data detection and separation for the mobile smart healthcare. Two steps were implemented in the study, namely a preprocessing step and a real-time processing step. PCA and Correlation Coefficient were used for feature selection and feature extraction. By this, the system can detect false physiological readings and can distinguish between the false and true medical functions. Other researchers (*Mohamed, Meddeb-Makhlouf & Fakhfakh, 2019*) attempted to improve the detection efficiency by proposing an intrusion cancelation

approach, thereby making the anomaly detection in medical devices efficient. This was achieved by using filters for eliminating noises in the medical data followed by detecting intrusions through statistically analyzed amplitude and frequency. Finally, the detected intrusions were removed to execute the anomaly detection in the medical device for diagnostic purpose. The simulation results applied on two sensor data showed that the system has high TPR and comparable FPR.

Anomaly or malfunction sensor reading resulted from defective sensor nodes or produced by fraudulent foreign entities can contribute to medical error and even mortality in patients. Researchers (*Salem et al., 2014*) suggested a system to identify abnormalities in Wireless Body Area Network (WBAN) for pervasive patient and health surveillance. The proposed software combines advanced data mining and deep learning technologies with existing sensor fusion techniques. The suggested model uses Support Vector Machine (SVM) to identify irregular incidents of the received sensor data. When an anomaly is detected, a regularly updated and regressive prediction model is used to differentiate between the stable and faulty readings resulting in a higher TPR and lower FNR.

It was recently (*Hau & Lupu, 2019*) emphasized that identification of fake data injections in low dense Wireless Sensor Networks is important for maintaining the integrity of data, particularly in medical IoT systems. Hence, researchers proposed a framework for detecting false data using anomalies in temporal-attribute correlations between sensor measurements. This method is successful in detecting attacks when more than one sensor is colluding to record coherent measurements (*Hau & Lupu, 2019*). Nevertheless, with an increase in the number of colluding sensors, the detection efficiency degrades to a point that the detection fails when most of the sensors are colluding.

Furthermore, researchers (*Nagdeo & Mahapatro, 2019*) have implemented an ML model to separate anomalous data from legitimate sensed data. This research used a combination of ANN with Ensemble LinReg as a detection technique for abnormalities in WBAN sensors. Firstly, normal, and abnormal health records were classified. Secondly, regression methodology was used to recognize the anomaly and real vital data. For the validation purpose, real medical physiological datasets were used. It was concluded that the system was able to effectively detect the anomaly.

Additionally, measured data of blood glucose sensor was examined in order to detect adversarial and accidental data modification intrusions (*Verner & Butvinik, 2017*). Here, Otsu's thresholding algorithm was used with extra statistical analysis to create different informative feature vectors. Then, linear SVM with different misclassification parameters were used to classify the feature vectors. The obtained results on huge patient's data showed 100% precision and 99.22% recall. This demonstrated that data modification attacks can be effectively detected by utilizing the ML approach.

Furthermore, other research groups (*Kintzlinger et al., 2020*) presented CardiWall system, which is used for detecting and preventing resource depletion attacks against ICDs at programmer device. The system is a collection of six security layers, exploitation of health experts understanding, statistical techniques and one class SVM as the ML technique. To perform the assessment, data were collected over a time span of 4 years and 775 benevolent clinical commands were used. These were belonging to hundreds of

specific patients and 28 malicious clinical prompts established by two cardiac specialists. The evaluation results showed that only two out of the six layers proposed in CardiWall system have provided a high detection capability. One class SVM failed to obtain a high detection rate because of the problems related to the datasets and a smaller number of features related to the malicious data. Noticeably, the system achieved an AUC of 94.7% with a true positive rate (TPR) of 91.4% and false positive rate (FPR) of 1%.

Rathore et al. (2019) proposed a Long Short-Term Memory, which is a type of recurrent neural network. It was used for predicting and forecasting different signal patterns of Deep Brain Stimulation (DBS). Usually, Rest Tremor Velocity (RTV) is analyzed to know the neurological disorders intensity. In this way, the authors analyzed RTV values for designing and training the neural network. Multiple attacks have been introduced in the DBS context to simulate and distinguish various attack techniques. To assess the performance of the algorithm in terms of accuracy and reliability, real and false observations were listed and estimated at the run time.

## Authentication and access control

In this section, results and analysis of the relevant papers on this topic are given, while the main features and limitations of the reported works are summarized in Table 9.

Rathore et al. (2018b) proposed a biometric verification methodology based on ECG, in which the Legendre polynomial extraction and MLP classifier were used for identification and authorization, aiming at securing the data, network, and application layer. The suggested methodology is the first effort to use ECG signals to exploit MLP in authentication. The results were verified on the ECG dataset and showed that it is possible to achieve 100% test accuracy with 5-degree coefficients when the authorized person is identified.

In a research made by Pirbhulal et al. (2019), a biometric protection system based on ML technique was introduced, in which the attributes for learning process were derived from ECG signals. Nevertheless, in the testing process, the user is authenticated by using the unique biometric EIs generated from the ECG and the polynomial approximation coefficients. It was concluded that the proposed system can be used for real-time healthcare implementations.

In another work carried out by Mawgoud, Karadawy & Tawfik (2019), an authentication approach was proposed, which is based on SVM. The authentication was established by incorporating both trust management and SVM at the gateway to recognize the frequency of resource-constraint devices and the timing of access. The proposed approach acts to identify several IoMT sensor artifacts based on their pseudo-random exposure in both frequency and time domains. If the value of the time or frequency domain is the same as its distinctive Pseudo-Random Binary Sequence (PRBS), the device will be authenticated by the gateway, in which an average trust value of 1.9 and less than 0.5 value are considered as adversary. Results proved that the method is viable for interacting with the IoMT devices.

Furthermore, a study was performed by Barros et al. (2019) aiming at reducing the computational cost, in which the features were extracted from ECG signal by utilizing only

**Table 9 The summary of the studies reported on authentication and access control.**

| References | Methods | Good features | Gaps and limitation | Tools | Data info |
|---|---|---|---|---|---|
| *Rathore et al. (2018b)* | Legendre approximation and MLP | -High testing accuracy | -High computational overhead for IMD<br>-Not able to detect unknown attacks | -MATLAB<br>-Keras<br>-Theano | ECG-ID dataset |
| *Pirbhulal et al. (2019)* | MLP | -Efficient<br>-Light weight | -Evaluation not given<br>-Lacks validation metrics<br>-MLP is memory intensive | Not given | Not given |
| *Mawgoud, Karadawy & Tawfik (2019)* | SVM and trust management | -Low resource consumption<br>-Lower detection time | -Performance overhead not calculated<br>-Lacks measuring the cryptography security strength of the algorithm | Not given | Not available |
| *Barros et al. (2019)* | MLP, RF, SVM and Naïve bayes | -Low complexity<br>-High accuracy | -Performance overhead not calculated<br>-High Training overhead<br>-Reduced accuracy due to feature selection<br>-Not easy to implement with different sensors | Weka | Stress Recognition in Automobile Drivers database |
| *Zhang et al. (2018)* | 2DPCA, LDA, and MapReduce | -Improved accuracy<br>-Improved efficiency | -Performance overhead not calculated<br>-High training overhead<br>-Not suitable for tiny IoT devices | Hadoop | MIT-BIH Database |
| *Shang & Wu (2019)* | LOF model | -Improved accuracy in acceptance and rejection | -Not accurate when user's behavior not stable<br>-Different performance on different brands of smartwatch<br>-Sensors should be attached to body tightly<br>-Performance overhead not calculated | Samsung smartwatchTizen OS 3.0 | Self-created |
| *Musale et al. (2019)* | Statistical filers, RF, KNN, and MLP | -Lightweight<br>-Higher accuracy<br>-User-friendly<br>-Easily deployable | -Performance overhead not calculated<br>-No attack model is considered for gait authentication<br>-Environment can affect its accuracy<br>-Authentication fails if the user is far from smart home<br>-Its accuracy decreases in the case of increasing the users | -Motorola (smartphone)<br>-Python | Self-created |
| *Vhaduri & Poellabauer (2019)* | A combination of filters ((KS)-test, PC, SD based filter) and SVM | -Use of hybrid biometric.<br>-High accuracy<br>-Low error rate | -Degradation of Non-Sedentary performance for highly active periods<br>-Need for retraining<br>-Cannot detect online attacks<br>-performance overhead not calculated | MATLAB | NetHealth Study Dataset |
| *Mohsen, Ying & Nayak (2019)* | Cryptography, convolutional Neural network (CNN) | -End to end security<br>-Strong against multiple attacks<br>-Lightweight<br>-Real time | -Communication overhead is still high for medical devices<br>-High training overhead<br>-Performance overhead not calculated<br>-Some attacks may still occur in that 30 min of monitoring gap ex; sensor may be relocated and returned | Not available | Not available |
| *Punithavathi et al. (2019)* | a lightweight random projection technique | -Low complexity<br>-Lightweight<br>-High recognition accuracy | -Rejection rate not calculated<br>-Memory, energy usage is not calculated<br>-Missing attack validation | OpenCV 3.6 in Python 3.4. | DB1 and DB2 of FVC2002 and FVC2004 |

| References | Methods | Good features | Gaps and limitation | Tools | Data info |
|---|---|---|---|---|---|
| *Rathore et al. (2018a)* | Dynamic Time Warping (DTW) | -Lightweight<br>-Efficient<br>-High accuracy<br>-Less complex | -Missing validation against attacks<br>-Acceptance and Rejection rate not calculated<br>-Heavy for implantable devices | -LabVIEW<br>-MathScript RT Module | ECG-ID dataset |
| *Mohsin et al. (2019)* | PSO, Blockchain, AES | -Higher secure transmission<br>-Strong against spoofing and brute-force attacks<br>-High accuracy | -Key distribution in AES adds extra load on the channels<br>-Block chain is a heavy and resource extensive | Not given | MMCBNU_6000 database |

Note:
 Each row represents each paper under category authentication and access control, and the columns show the characteristics that are used for evaluating them.

fiduciary points measured from the acquisition of the signal. A set of ML techniques such as MLP, RF, SVM and NB were used for the classification purpose. The ECG signal of Stress Recognition Database was employed and it was proved that the method can be used for continuous authentication with lower complexity using less than 10 features.

Hospital systems with huge number of patients require proper authentication method. For that purpose, *Zhang et al. (2018)* proposed a hybrid technique that combines fiducial and non-fiducial features for accuracy improvement for authentication in healthcare systems. Feature selection of 2DPCA was used to enhance the accuracy of Linear Discriminant Analysis (LDA) and improving authentication of smart healthcare along with MapReduce. The PQRST peaks were selected as the main fiducial features. For the ECG user recognition, an incrementing training of the LDA algorithm was considered, utilizing the exposed fiducial and non-fiducial features. The authors reported that detection based on fiduciary feature and Fast Fourier Transform has provided a poor precision. Additionally, they concerned that growing the number of features would lead to additional computing effort.

Referring back to the individual authentication for daily use of wearable devices, researchers (*Shang & Wu, 2019*) used photoplethysmography (PPG) signals inspired by hand movements for user authentication on smartwatches. To protect from attackers, the software first senses the beginning and stopping point of any new hand gesture and decides if an attacker is present. This was realized by comparing the features derived from the observed hand movements with those obtained from the regular user. The training samples of the usual consumer have been used to create a local outlier factor (LOF) model. The LOF model has the feature of few parameter modification which helps to easily create a new classifier for a new consumer. In addition, LOF can identify anomalies efficiently. This approach resulted in an overall authentication accuracy of 96:31% and an overall true rejection rate of at least 91:64% against two forms of attacks. Authors claimed that this software-based solution can be integrated into any smartwatch with PPG sensors for multi-factor authentication.

On the other hand, *Musale et al. (2019)* proposed a lightweight authentication approach using gait biometric for authenticating users of commercial smart watches. This was

realized by extracting the statistical features and person's actions-related features from the collected data of sensors, thereby improving both accuracy and efficiency. For this purpose, techniques such as filters with RF, K-NN and MLP classifiers were used. Results demonstrated that the system could achieve a higher accuracy by having a reduced feature and fewer sensed dataset. This in turn renders the theoretical architecture more realistic and easily accessible to wearable IoT devices with minimal processing resources and energy efficiency.

Furthermore, in a study reported by *Vhaduri & Poellabauer (2019)*, a hybrid approach was implemented for biometric authentication in wearable devices. This was achieved by utilizing a three-combination form of course-grain minute-level biometrics: behavioral (step counts), physiological (heart rate), and hybrid (calorie burn and metabolic equivalent of task). An analysis was performed by using 400 Fitbit consumers data that were obtained from a health research collected over 17 months. A combination of filters (KS-test, PC and SD) with SVM of different settings were used. A high accuracy for both sedentary and non-sedentary with low error rates of 0.05 were achieved when a binary SVM was used. It was also proved that hybridizing the biometrics gives better results even during non-sedentary periods.

*Mohsen, Ying & Nayak (2019)* proposed ECC defined lightweight cooperative authentication scheme to be used in real-time medical wireless sensor networks among physicians/nursing staff, trustworthy databases, sensors and patients. Here, the doctor/nurse can use the system by his/her fingerprint, while patient identity can be verified through continuous monitoring of physiological data (e.g., ECG signals) in every 30 min to detect the physical theft of the sensor. The server uses PID and ECC feature set as training data and utilizes convolutional neural network. Dynamic identity was applied for user anonymity and mitigating user traceability. Results showed that the scheme is strong against multiple attacks with accepted performance.

Moreover, *Punithavathi et al. (2019)* proposed a lightweight approach for cloud cancelable biometric authentication. As such, privacy problems related to biometric utilization can be tackled by getting advantage of cancelable biometric templates. A random projection method was used for generating the cancelable fingerprint template. Fingerprint images were chosen anonymously from publicly accessible databases. After pre-processing and feature extraction, the lightweight random projection method was used to create cancelable fingerprint models. However, authors mentioned that the work's scalability in real world has not been tested.

*Rathore et al. (2018a)* implemented a lightweight approach for trusted authentication using Dynamic Time Warping (DTW) method. A large database from Physionet was used for validating the method and informative features were extracted from ECG using DTW. It was seen that DTW method showed a higher accuracy compared with that of the non-linear SVM. Additionally, it took less time, while both methods have the same memory complexity.

Moreover, *Mohsin et al. (2019)* designed a secure framework for user verification in two stages of patient authentication. In the first stage, merged RFID and finger vein (FV) features were generated to increase the security levels. In the second phase, a combined

technique of AES encryption, blockchain, and PSO steganography was used for secure transmission of the data. In the evaluation process, 106 samples were chosen from a database comprising of 6,000 samples of FV images. It was seen that the system is strong against spoofing and brute-force attacks, whereby an improvement of 55.56% was achieved in the secure biometric transmission.

## Intrusion and malware detection

Herein, the main achieved results on this topic that were reported in literature are given followed by a summarized comparison of the studies, as shown in Table 10.

*Begli, Derakhshan & Karimipour (2019)* designed a security framework for smart healthcare system, which can be specifically useful for protecting wireless sensor network against unauthorized access and network attacks. In their work, a multiagent based layered architecture was first defined. Then, the IDS was applied using SVM, which is proportional to the level of energy and the sensitivity of the available data of each triple group of the agents. Different rules were used for each layer of the multiagent healthcare. Also, attacks related to healthcare, including eavesdropping attacks, were studied in terms of energy usage and computational cost.

Moreover, a multivariate correlation analysis was proposed for the IoMT attacks detection (*Itten & Vadakkumcheril, 2016*). Triangular based area maps were utilized for analyzing the incoming data's features to formulate a correlation among various features. It was seen that the learning model has a high rate of time usage with increased classification accuracy. The proposed method can differentiate both known and unknown DoS attacks from normal network traffic.

Additionally, in a study made by *Schneble & Thamilarasu (2019a)*, a massively distributed intrusion detection based on ML was designed and implemented for the Medical Cyber Physical System. Primarily, the notion of Federated Learning was researched to reduce the connectivity and computing expenses associated with conventional ML algorithms. Evaluation was carried out using real patient records against the attacks including DoS, data manipulation and false data injection. Observational findings showed that the proposed system attained a high accuracy of 99% and FPR of 1% including a decreased communication network's overhead. Moreover, they proved that the program could cope with unevenly distributed data and is a flexible approach that utilizes the computational power of multiple mobile devices. In another work (*Odesile & Thamilarasu, 2017*), the authors have introduced a mobile agent-based intrusion detection program for WBAN. Different types of sensor agents, and cluster agents are used. Multiple threats were detected in a distributed and hierarchical framework within the healthcare network. ML methods such as NBC, KNN, SVM, RF and DT were applied to sensor nodes for the purpose of providing precise attack detection, followed by choosing an appropriate approach. The system has been tested to be effective in terms of precision and power usage. It was revealed that there is a lack of sufficient researches on the intrusion detection in the medical cyber-physical system. Since real time attack detection is critical for medical devices, the volume of data analyzed by the IDS should be reduced to maintain optimal frequency detection. Therefore, an effective IDS was

**Table 10 Summary of the studies reported on intrusion and malware detection.**

| Ref. | Method | Type of intrusion detection | Good features | Gaps and limitations | Tools (software & hardware) | Datasets |
|---|---|---|---|---|---|---|
| *Begli, Derakhshan & Karimipour (2019)* | SVM | Anomaly and signature-based IDS | -High detection accuracy <br> -Satisfied detection time | -High memory overhead <br> -Low detection rate in misuse analysis <br> -Outdated dataset | R program | NSL-KDD |
| *Itten & Vadakkumcheril (2016)* | a multivariate correlation analysis | Hybrid Anomaly and rule based IDS | -Improved accuracy <br> -Detecting known and unknown attacks | -High learning overhead <br> -Unclear implementation <br> -Validation process not given | Not given | Not given |
| *Schneble & Thamilarasu (2019a)* | Federated Learning | Anomaly based false data injection, data modification, DoS IDS | -Multiple attack detection <br> -High accuracy <br> -Low FPR <br> -Flexible | -Reduced accuracy and increased FPR in some cases <br> -Weak against adversarial attacks | -Sci-kit Learn's on Raspberry Pi's <br> -MATLAB | -MIMIC dataset from PhysioNet <br> -ECG |
| *Odesile & Thamilarasu (2017)* | Hierarchical and distributed classifiers (NBC, KNN, SVM, RF and DT) | Anomaly based IDS | -High accuracy <br> -Low energy consumption | -High training time for some methods <br> -High FPR | Castalia WBAN | Self-created simulated data |
| *Schneble & Thamilarasu (2019b)* | feature selection (Laplacian scoring) | Signature based intrusion detection. | -Reduced the detection time <br> -Reduced performance overhead | -Accuracy reduced upon selecting more features <br> -Lacks detail of selected features | Not given | MIT-BIH Arrhythmia dataset |
| *He et al. (2019)* | SAE is used for feature selection | Signature based intrusion detection HIDS | -Improved accuracy and detection overhead | -High training overhead <br> -Not lightweight | Not given | real CHS Source not given |
| *Swarna Priya et al. (2020)* | Feature selection using PCA and GWO with DL classifier | Signature based IDS NIDS | -High accuracy <br> -Low training and testing time | -High Memory and CPU overhead <br> -Limited to only IP based devices | Not given | NSL-KDD |
| *Thamilarasu (2016)* | multi-objective genetic algorithm (GA) | -Signature based IDS <br> -HIDS | -Feature selection reduced the complexity | -Not lightweight <br> -Detection time and training time not given | MATLAB | Self-created by simulation |
| *Asfaw et al. (2010)* | Data mining-based Association rule mining | -Anomaly based HIDS | -Ability to detect anomalous activity. | -Lacks model evaluation <br> -Using many resources | -J2ME <br> -Java Servlets <br> -MySQL Server <br> -Palm PDA | Self-created |
| *Manimurugan et al. (2020)* | Deep Belief Network (DBN) | -Anomaly based NIDS | -High accuracy <br> -High precision, F1, and recall | -FPR and performance overhead neglected <br> -High training overhead | MATLAB | CICIDS2017 dataset |

| Ref. | Method | Type of intrusion detection | Good features | Gaps and limitations | Tools (software & hardware) | Datasets |
|---|---|---|---|---|---|---|
| *Alrashdi et al. (2019)* | Ensemble an online sequential extreme learning machine (EOS-ELM) | -Anomaly based NIDS | -Lower latency compared with cloud-based<br>-Reduced attack detection time | -Memory and CPU usage are not considered | -Python (scikit-learn, Tensorflow, Keras, Numpy, HDF5) | NSL-KDD dataset |
| *Wazid et al. (2019)* | Machine Learning, Data mining, Blockchain | -Signature and Anomaly based Malware detection | Discussed current IoMT malware detection | Most papers are generic IoT especially Smart phone-based solutions. | Not given | Not given |
| *Fernandez Maimo et al. (2019)* | NFV/SDN, OC-SVM, and Naïve Bayes | -Anomaly and signature-based Ransomware detection | -Real time<br>-Can detect recent malware attacks<br>-Short detection time | -Not lightweight<br>-Performance overhead not calculated | -OpenICE<br>-OpenStack<br>-OpenDaylight<br>-Python language<br>-Scikit-learn v0.20.0 | Self-created real testbed dataset |
| *Shakeel et al. (2018)* | Deep-Q-Network (LDQN) | -Signature based malware detection | -High detection rate<br>-Low error<br>-Low energy consumption | -FPR, Memory usage are not considered<br>-Training overhead is high | NS2 simulator | Simulated data |
| *Landau et al. (2020)* | Different ML techniques | -Privacy attack detection | -Improved performance<br>-Larger datasets used | -Still low accuracy<br>-High error rare | Not given | rsEEG data |

**Note:**
Each row represents a paper under category intrusion and malware detection, and the columns show the characteristics that are used for evaluating them.

proposed by addressing the problem of feature selection to reduce the data dimensionality. The findings showed that Laplacian scoring strategies are effective in optimizing the collection of features with reduced resource usage.

Furthermore, for the detection of anomaly based intrusion, a stacked autoencoder (SAE) was proposed by *He et al. (2019)*. The SAE was used to extract more informative features and eliminate feature dimensions, which resulted in the reduction of detection overhead.

*Swarna Priya et al. (2020)* applied a hybrid PCA-GWO algorithm for feature selection and DNN classifier for classifying the network attacks. The proposed methodology suits the IoMT devices that are using a unique IP. One-Hot encoding scheme was used for pre-processing the input data. Then, PCA and GWO algorithms were sequentially utilized for further data reduction followed by the use of well-known classifiers for prediction. Results showed that the hybrid PCA-GWO is capable of increasing the detection accuracy by 15%. In addition, the training and classification time was reduced by 32%.

For the same purpose, researchers (*Thamilarasu, 2016*) used a multi-objective GA algorithm for the feature selection of WBAN network attack detection. The experimental

results showed that the proposed algorithm could include the useful features for detecting a specific attack in the detection process, thereby decreasing the computational complexity.

*Asfaw et al. (2010)* presented a datamining-based model to provide a host-based anomaly and attack detection method for pervasive healthcare systems. When a mobile is requesting from the server, its message is recorded and fed to the classification model. Then, the model classifies the record as benign or malicious depending on the previously recorded history. Eventually, the classifier holds the reactions in a passive fashion, maintaining each specific record with the disruptive behavior and hence detecting the anomalous events.

In a recent work performed by *Manimurugan et al. (2020),* Deep Belief Network (DBN) was proposed for attacks detection in the IoMT. The measurement criteria used in the study were precision, recall, accuracy, and F1-score. The suggested model achieved positive results across all variables compared to the other techniques. It was claimed that this model can be expanded for detecting several forms of attacks against IoT devices and different databases.

*Alrashdi et al. (2019)* presented a fog-based attack detection (FBAD) architecture by utilizing an online sequential extreme learning machine (EOS-ELM) collection for monitoring of suspicious behaviors in healthcare system. They proved that the proposed architecture is effectively implemented in the decentralized fog-attack detection by comparing its efficiency to other methods. It was revealed that the decentralized architecture surpassed the centralized framework in terms of detection time and accuracy of classification.

*Wazid et al. (2019)* surveyed the malware detection methods in the IoMT network using ML approaches. They elaborated on how serious the malware attacks are, especially botnet attacks on the three tiers of security and privacy. In the presence of such attacks, the sensitive data of IoT communication may be disclosed, altered or even may not be available to the authorized users. Hence, in the study various types of malware attacks were explored with their symptoms and a taxonomy for the IoMT security was given. Moreover, ML based malware detection methods were discussed.

Furthermore, *Fernandez Maimo et al. (2019)* used ML techniques for detecting and classifying ransomware attacks in ICE. The NFV/SDN methods were used to isolate and remove contaminated medical equipment and networks. The method was developed to detect recent malwares such as WannaCry, Petya, BadRabbit and PowerGhost. Techniques such as OC-SVM and Naive Bayes have been proved to detect and classify ransomware infecting ICE with respective accuracy and recall of 92.32% and 99.97% for the OC-SVM in anomaly detection. This is where the Naive Bayes classifier was able to reach a classification accuracy of 99.99%.

*Shakeel et al. (2018)* studied a secure data access and transmission in the IoMT through utilizing the Deep-Q-Network (DQN) methodology. Initially, the IoMT system was analyzed by the deep neural network to authenticate and eliminate any possible malware attacks. The traffic attributes of each request were derived and recorded in the database. The output attribute was then analyzed by using the state feature and associated behavior. Then, the deep neural convolution network was used to classify them into

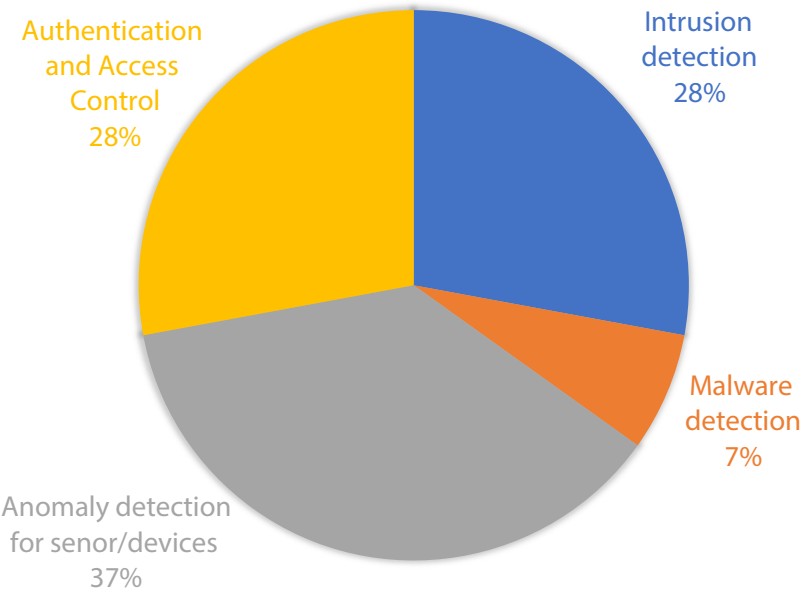

**Figure 7 Papers distribution by the direction and problem-solving domain.** The pie chart shows the direction of studies. The percentage of the papers for each direction of study is given in the chart.

malware and regular data. Results showed that the method has achieved a low error of about 0.12 with malware detection rate of 98.79%.

Furthermore, with the aim of protecting the brain activity data from attacks in BrainComputer Interface (BCI) systems such as EEG and fraud BCI program (e.g., game), which lets a malicious entity to obtain the user's brain activities, researchers demonstrated that a ML method can anticipate relevant identity characteristics by studying resting-state EEG (rsEEG) records of a person's brain activity (*Landau et al., 2020*). A complete collection of raw rsEEG tests along with dissociation degree and executive function (EF) assessment indicators were used for 162 subjects in the test. Their study concluded that breaching those identical brain activities are possible if proper security measures are not taken into consideration.

## DISCUSSION

In this section, the main outcomes of the reviewed studies are elaborated and discussed aiming at achieving fruitful answers to each of the research questions, respectively.

### RQ1: what is the current state of the art and direction of study in the IoMT security using ML?

One can see from Fig. 7 that 37% of the research articles were devoted to detect anomaly at sensors or medical devices. This indicated that in the past years, the main focus was on detecting the intrusions such as false data injection, resource depletion attacks, behavioral and physical attacks against medical devices. This can be attributed to the fact that these attacks are serious and they lead to significant health issues. For instance, it may lead

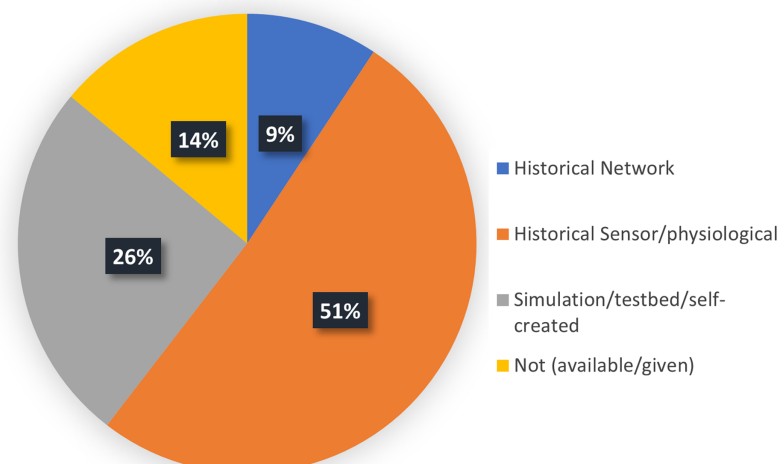

**Figure 8** **The type of data used by the researchers to conduct their research.** The pie chart shows the percentage of the papers in terms of the type of the data used for their analysis.

to death in the case of attacks to the implantable medical devices (*Alsubaei, Abuhussein & Shiva, 2019b*; *Hatzivasilis et al., 2019*; *Yaacoub et al., 2020*).

Also, intrusion and malware attack detections received a considerable attention, counting for 35% of the reported studies (7% for malware and 28% for other attacks). Nevertheless, the majority of the papers were found to have focused on NIDS. On the other hand, papers that used different strategies to solve the authentication and access control issues were found to take 28%. However, as authentication is computationally heavy for the IoMT devices, the current direction of research is to apply lightweight mechanisms (*Aghili et al., 2019a*; *Wu et al., 2018*) by using sensor physiological data to reduce the computation load on the device.

## RQ2: what kind of data and tools are used for applying ML techniques in the IoMT security?

Machine Learning methods are data dependent as they learn from these data overtime and decide intelligently based on their learning ability, amount and quality of the data. For this reason, in our study we have paid attention to this matter and we have analyzed all the selected research articles based on the type of data they used for decision making and learning process. It can be seen from Fig. 8 that most of the papers have used historical benchmark data, of which 9% have used network data, while 51% have used sensors and physiological data. This is mainly because most of the methods were to find anomalies in the sensors and to use device authentication as security solution. On the other hand, 26% of the papers have used simulated or emulated data. However, some of the studies have not given the source of their data or did not mention it at all. For this category, we have given the label not available or not given. This group of papers provides 14% of the whole selected papers.

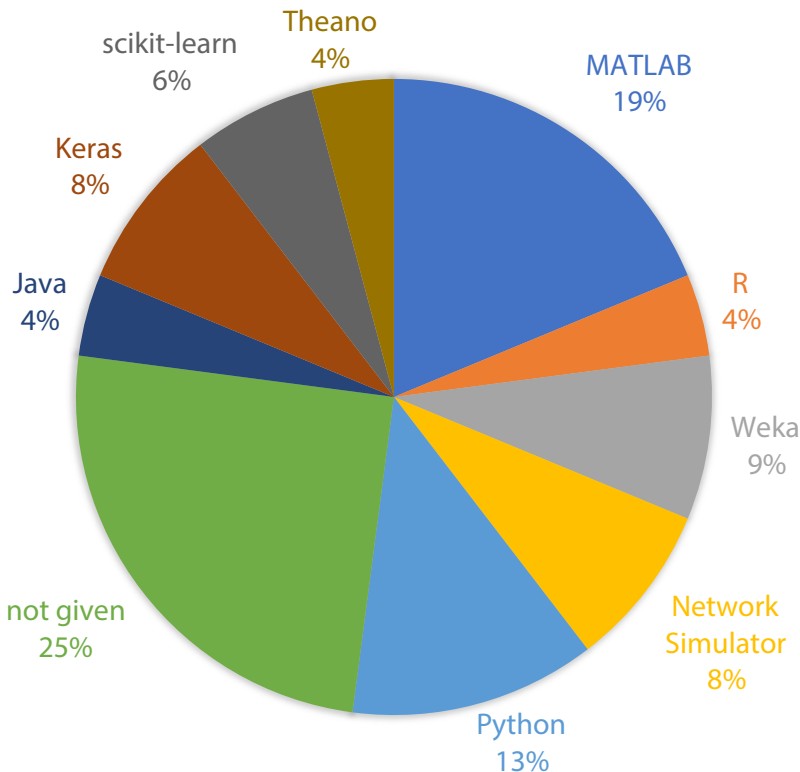

**Figure 9 Software and tools used by the studies.** The pie chart shows the percentage of different tools and software which were used by the reviewed studies.

  To further answer the above research question, software and hardware tools used and reported in the reviewed studies are analyzed. Figure 9 shows the used software tools and programs in the reviewed studies. Tools used by the studied works were mainly Network/Sensor Simulators with ML tools. However, we have excluded those studies that used simulation but did not mention its tool. For this reason, the percentage of the simulation tools is less than expected. Furthermore, 25% of the studies did not give the tools used in their studies. The remaining were mainly programing tools for ML purpose. We can see that MATLAB has been used more than the rest of the tools. Python also gained good attention in the current years. In addition, Weka tool has been used frequently, which counts for 9% of the studies. Additionally, Keras and Sckit-learn libraries were used with 8% and 6% by the studies, respectively. Those represent open-source libraries are usually used with Python. Moreover, in the reviewed studies, some works have used testbeds and hardware tools, while few of them reported their tools. Figure 10 shows the number of those tools that have been used. It was found that six types of devices and hardware tools were used and reported in the studies.

### RQ3: how ML techniques are effectively applied by the studies and what are their limitations?

Through a critical analysis of the current works, we can conclude that traditional ML techniques may fail if proper considerations are not given to some metrics such as

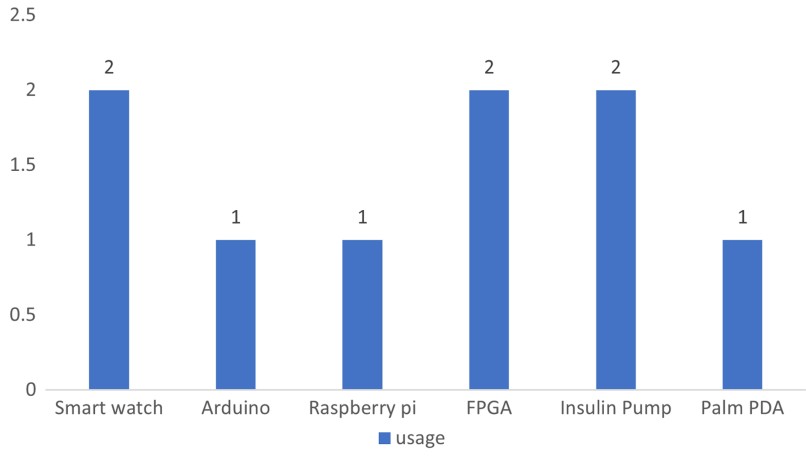

**Figure 10 Devices and hardware tools used in the studies.** The bar chart shows the number of hardware tools used by the analyzed studies.

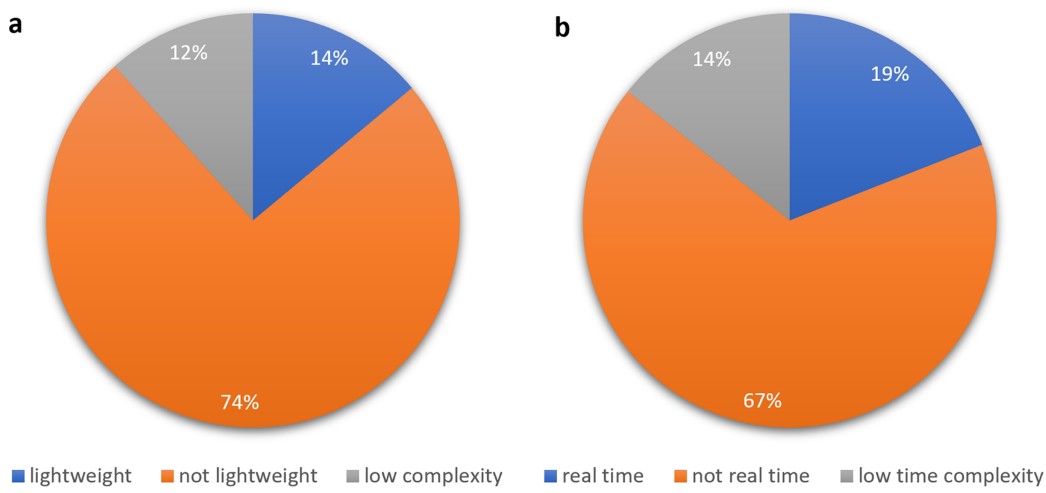

**Figure 11 Analysis of the studies in terms of (A) complexity and (B) real time analysis.** The pie chart (A) shows the percentage of the papers in terms of complexity {lightweight, heavy, low complex} The pie chart (B) shows the percentage of the analyzed papers based on their real time feature {real time, not real time (heavy), low time complex}.

computational complexity and energy usage. ML performance is reduced when few data are used. However, the IoMT devices are resource limited such as (IMDs) and using a huge data analysis on these devices is resulted in degrading their lifespan. For these reasons, one should take a balance between these two aspects. From our analysis of the reviewed papers, we have found that a small portion of the studies have used lightweight approaches. The pie chart shown in Fig. 11A illustrates that only 14% of the studies have used lightweight approach. Additionally, 12% of the studies have used low complexity approaches, that means it is not exactly lightweight but not heavy. The rest of the studies either did not require this feature in their system or failed to apply it. Furthermore, real time ML techniques are crucial for the IoMT, especially in attack and intrusion detection. Because the IoMT networks are dealing with streaming data and some of the

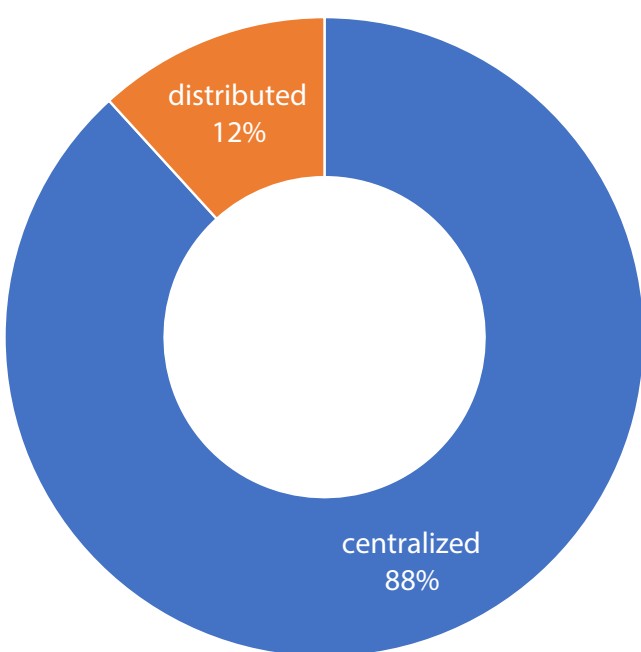

**Figure 12 Details of the studies in terms of placement.** The chart shows the percentage of the papers based on their placement.                 

devices never stop from working, a continuous and fast detection of attacks are required. It is seen from Fig. 11B that 19% of the studies were used real time approach, while 12% of the studies have used low complex approaches that are not exactly real time but have approached it. Nevertheless, the other 67% of the works were offline. Additionally, the studies focused on network attack detection paid less attention to distributed attack detection. As can be seen in Fig. 12 that only 12% of the studied have used decentralized models, while the rest were centralized. However, a hybrid of the methods was considered as a better approach.

Based on the studies gaps that were assigned previously, it is found that majority of the studies have focused on improving the traditional evaluation metrics for their ML models such as high (accuracy, recall, precision) and low (FP, FN). However, performance metrics such as memory (space), CPU, time, and energy overhead were neglected. Additionally, most of the techniques of attack detection were performed in an isolated offline environment. However, one should consider simulation/hardware implementation to represent a real IoT setting. Also, decentralized and hybrid approaches are much compatible for the IoMT than centralized one, which was again not common among the studies. Furthermore, we have noticed that the datasets used by the attack and intrusion detection methods were out of date and some of them do not represent the current IoMT system. Moreover, sensitive, and private data usage should be avoided in attack and anomaly detection tasks which was also given a minor attention. Therefore, there should be a trade-off among preserving privacy, high accuracy, and computational complexity. Another challenge that should be considered is the risk of adversarial attacks on ML techniques themselves. An attacker who has enough knowledge on how the

ML techniques work with the data can manipulate the data at training or testing stage to fool the ML method. Hence, the proposed methods should be strong enough against the adversarial attacks.

### Limitations

The underlying drawbacks of this research are: (i) Selection bias: This study focuses on literature research, which may have unintentionally omitted most recent non-scholarly advancements or scholarly articles that have not yet been published at the time of preparing this study. (ii) Publication bias: Some of the studies may have been discarded since their full paper was not accessible. (iii) Descriptive bias: Despite the efforts made to present the background comprehensively, this SLR is combining many topics that may need someone to refer to the external sources for a deep understanding.

## CONCLUSIONS

In this study, a comprehensive Systematic Literature Review (SLT) was given about the IoMT security and privacy issues and how Machine Learning (ML) methods are used for solving them. By examining the content of the study, including methods, good features, limitations, tools, and datasets, the designated research questions were answered. Findings of this study showed that ML techniques are effective in addressing the IoMT security issues with promising results. Majority of the studies was devoted to device layer or body area network security since attacks on devices such as IMDs are seriously affecting the patient's health and life. The security solutions for such devices were sensor anomaly detection and device authentication and access control. Furthermore, securing the network layer was seen among the studies that used attack and malware detection strategies.

Moreover, the tools and environment of the current works are a combination of network simulators and ML tools with more focus on the latter. Additionally, there is a lack of relevant datasets, especially in the intrusion detection. Most of the studies focused on improving the common ML algorithms evaluation metrics such as high accuracy and low FPR. However, since the IoMT devices are characterized with shortage in power and small memory and processor, there should be a balance between security and maintaining resources lifespan during the adaption of these solutions. We have concluded that traditional ML techniques may fail if proper consideration is not given to some metrics such as resource complexity, time complexity, and energy usage. It was noticed that a vast majority of the studies ignored these criteria in the evaluation of their proposed models. Therefore, ML techniques are vital in the application of the IoMT security. However, future studies should focus on how to use ML in a proper way to concede the nature of the IoMT.

### Funding

The authors received no funding for this work.

## Competing Interests

The authors declare that they have no competing interests.

## Author Contributions

- Shilan S. Hameed conceived and designed the experiments, performed the experiments, analyzed the data, performed the computation work, prepared figures and/or tables, authored or reviewed drafts of the paper, critical analysis of the reviewed papers based on their methods, strength, and limitations, and approved the final draft.
- Wan Haslina Hassan conceived and designed the experiments, analyzed the data, authored or reviewed drafts of the paper, and approved the final draft.
- Liza Abdul Latiff conceived and designed the experiments, authored or reviewed drafts of the paper, and approved the final draft.
- Fahad Ghabban conceived and designed the experiments, authored or reviewed drafts of the paper, and approved the final draft.

## Data Availability

This is a review article; all related data are available in the Results and Discussion sections.

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
