# Peer review of "A systematic review of security and privacy issues in the internet of medical things; the role of machine learning approaches"

_PeerJ Computer Science, doi:10.7717/peerj-cs.414_

## Round 0.1 · original submission · Major Revisions

The article is seen as a useful and welcome contribution in the area of security/privacy and health. Yet, they identified a number of issues that make the article not ready for publication. My summary of the main points is:
- The presentation overall needs to be improved. The motivation needs to be better explained, the article better structured and the separate sections need to be better connected.
- The findings should be better discussed in the context health care.
- The study design needs to be described in more detail to make it replicable.
- The threats to the validity of the findings need to be discussed in more detail.

Reviewer 1 ·

Basic reporting

This is a review paper on medical Internet of things. The author lists and analyzes the related research work on the security and privacy protection of medical Internet of things, but it still needs further analysis and summary, and lacks of refinement. Some spelling mistakes in the paper also need further research

Experimental design

This is a review paper on Internet of medical things.

Validity of the findings

This is a review paper on Internet of medical things.

Additional comments

.

Reviewer 2 ·

Basic reporting

the major issue that needs to be addressed, to have a better overview of the problem and a more complete paper, is the lack of a clear motivations for this work. It is mentioned how the IoMT is potentially exposed to cyber-attacks, and that ML can potentially solve this issue, but (i) the article should elaborate more on how and why the IoMT is specifically more exposed to such threats; (ii) what attacks can be made and why they would be problematic in the IoMT scenario; (iv) how ML can help solve them; (v) why is ML the best candidate for security and privacy in the IoMT, and why traditional security algorithms and countermeasures fail or are not taken into account (please, be specific and offer technical explanations or examples).

The article definitely needs to be revised, proofread, and corrected. The use of English is incorrect, and several sentences are difficult to understand or poorly written. Moreover, many typos are present in the paper especially in the first third.

See attached pdf for more details

Experimental design

The organization of the article is confusing, especially in the initial part (at least till page 16, line 412), and results difficult to follow for the reader. Having a clear overview of the problem, as stated above, would improve this aspect. Second, the other reason why it is confusing is because the first sections mixes together definitions, general aspects of IoT, IoMT, ML and details on security and privacy. The effort to make the paper readable by non-experts as well is noticeable, but there should be a better linking between sections and among paragraphs in each section.

Validity of the findings

As for the discussion at the end, it does not offer a good insight into how the techniques analyzed in the previous sections can be applied to the healthcare field, or what the challenges are. They should be connected better with the previous part of the paper.

Additional comments

see attached pdf

Annotated reviews are not available for download in order to protect the identity of reviewers who chose to remain anonymous.

Reviewer 3 ·

Basic reporting

This article presents the design, execution and results of a SLR of (1) the use of using machine learning approaches (2) to address security and privacy issues (3) in IoMT.

* Clear and unambiguous, professional English used throughout.

The language is clear and mostly correct, with minor typos: (l.89) “rational” instead of rationale, (l.125) “principle tiers”, (T.5) “Does the paper is…”.

* Literature references, sufficient field background/context provided.

The references in introductory material are adequate.

* The article should include sufficient introduction and background to demonstrate how the work fits into the broader field of knowledge. Professional article structure, figures, tables. Raw data shared.

The abstract devotes too much length to motivation and none to the survey method (sources? filters?), execution (how many papers? How many filtered out?), and results (it only reports an observation without clear relation to the examined literature).
The Research Rationale (not Rational) proposes research questions left unnamed, but they are later referenced as RQ1 etc.
The Background section describes (1) several ways of organizing IoMT architectures and types of sensors; (2) security and privacy issues in IoMT (although confusing metrics with quality characteristics, e.g. Privacy); (3) machine learning and some uses in IoT; and (4) deep learning and big data in IoT security
The Survey Methodology sections is too insufficiente (see below).
The Results and Findings presents detailed demographic and approach results, but the latter are not organized in a coherent single schema (see below).
The Discussion summarizes all primary studies, but is organized in non-orthogonal categories.
Fig.1 seems unnecessary (in style and content).

* Is the review of broad and cross-disciplinary interest and within the scope of the journal?

The review addresses a topic at the intersection of security/privacy, machine learning and IoMT, all three topics that are within the journal scope; the topic and intersection are explained in the Introduction. The target audience is explicited in Research Rationale

* Has the field been reviewed recently? If so, is there a good reason for this review (different point of view, accessible to a different audience, etc.)?

There are recent surveys of IoT security and privacy, but none of machine learning techniques applied to this problem. The problem is narrow, but exploding on terms of published work, and deserves a specialized review.
Related work is reviewed but somewhat carelessly: (l.298) “Yang et al (Cui et al 2018)”, (l.311) “Sikder et al. (Newaz et al. 2020)”.

* Does the Introduction adequately introduce the subject and make it clear who the audience is/what the motivation is?

The Introduction explains the three survey topic elements: (1) IoMT, (2) privacy and security issues in #1, and (3) machine learning approaches to #2. The paper structure is given at the end of section, but it does not fit the actual sections titles/matters.

* Formal results should include clear definitions of all terms and theorems, and detailed proofs.

N/A

Experimental design

* Methods described with sufficient detail & information to replicate.

The first major problem in this survey is the lack of replicability: the overall method is well described, but needs to have specific databases searched, dates, specific query strings (the mind map in Fig.2 is suggestive but insufficient), indicate how many papers were found from each source, how many were duplicated, how many were filtered out at each step…
Also, most filtering criteria (Table 5) seems inadequate: #3 is always true given the searched databases; #4 makes initial filtering depending on the RQ, a methodological tail-biting; #5 is irrelevant (if it were, the search should be only recent papers).

* Is the Survey Methodology consistent with a comprehensive, unbiased coverage of the subject? If not, what is missing?

The survey methodology is fairly standard, but the authors seem unaware of its methodological moorings and consequent steps, risks, reporting criteria, etc.; please check “Guidelines for conducting systematic mapping studies in software engineering: An update” (Petersen et al. IST 2015) or any of Barbara Kitchenham’s papers.

* Are sources adequately cited? Quoted or paraphrased as appropriate?

The sources are mixed up with the papers found in the review; it is important to preserve the distinction.
Several sources are mis-cited, e.g. “Sikder et al. (Newaz et al. 2020)”, which would be correct iff Sikder == Newaz.

* Is the review organized logically into coherent paragraphs/subsections?

The demographic data is presented in aggregated form, and there is no way to ascertain trends (e.g. how the journal-conference balance has changed).
The second major problem in this survey are the detailed results: they are carefully presented, but organized into thematic sections that are themselves not motivated, i.e. there is no single schema that accounts for them. The survey introduced (in the Background) several IoMT architectures, several classifications of risks, and several kinds of machine learning… but this information is not used to organize the primary studies in some orthogonal classification. Thus, some papers would fit in several categories (e.g. because they address several problems or use several techniques), but that cannot be visualized from the way the results are presented, and a researcher focused on a category would miss a paper that was deemed to “better fit” in another category. A better presentation would be (for example) a matrix with all found papers indicating which categories they fit.
The Discussion of RQs is generally adequate, but the RQ4 is too shallow because the gathered data (organized by kind of attach etc.) is not useful for reasoning about quality characteristics (see “Security in Telehealth Systems From a Software Engineering Viewpoint: A Systematic Mapping Study” (Marquez et al Access 2020) for a survey organized along those lines).

Validity of the findings

* Is there a well developed and supported argument that meets the goals set out in the Introduction?

The study Limitations are not well addressed, and this is a key point in any systematic survey. Please identify and address internal (consistency) and external (generalizability) validity threats.

* Does the Conclusion identify unresolved questions / gaps / future directions?

The Conclusions should summarize the key findings, which are actually well discussed in the paper itself (when revising the RQs). Some of the most important implications, like the lack of datasets and the big block that this posits to machine learning, are mentioned in passing and without connecting back to actual primary studies. The conclusions need a thorough revamping.

Additional comments

This article is timely and welcome, and it reflects a large amount of careful work. However, its presentations has two problems that must be overcome:
* the methodological underpinnings of systematic reviews require greater formality in reporting, to allow reproducibility
* the results are organized in sections that do not reflect an overall classification, although several elements for such classification are proposed in the background section of the survey itself; as a result, non-narrow papers may end up recorded as fit to only one category

---

## Round 0.2 · accepted · Accept

Thank you very much for revising your article. All reviewers are now happy with the article in its current form.

Please include in your final version the comments by Reviewer 2 on the search strategy:
- selection of papers
- removal of papers

Reviewer 1 ·

Basic reporting

no comment

Experimental design

no comment

Validity of the findings

no comment

Additional comments

This paper proposed a systematic review of security and privacy issues in the Internet of medical things. The recent research papers of IoMTs are reviewed and a critical analys are conducted on the selected papers.

Reviewer 2 ·

Basic reporting

The authors have satisfactorily addressed the concerns and questions of the reviewers.
The outcome is a well written, clearly organized and easy to read and easy to follow article.
The motivations of the work are clear in the revised version of the article.

Most of the grammatical errors and technical incorrect text of the first version have been fixed.

Few typos still remains, and I'm listing some of them. I recommend "minor revisions" just to correct these mistakes and address a couple of minor points reported below. However, the paper is in good shape for publication and would not necessarily need to be re-reviewed.

Experimental design

The article is in Aims and Scope of the journal. The investigation performed and methodology are technically sound and objective.

One point the authors might want to clarify is relative to the search strategy used for selecting the research papers (Figure 2), and specifically how the 27 papers were removed at the end based on "abstract, conclusions, quality of paper".
It is clear that the criteria used are those in Table 6, however, citing a criterium such as "quality of paper" is confusing because it is a subjective one. The wording can be left without changes, but I suggest clarifying in the text what it exactly means.

The review is logically organized into coherent paragraphs/subsections.

Validity of the findings

The authors have improved the result discussion, and included tables and figures that help understand and visualize the outcome of their research.

Additional comments

The article has improved, and it was nice to read. However, here are some remaining typos that the authors might want to correct:

Line 34: a critical analysis were -> was
Line 62: body sensor -> body sensors
Line 63: (gateway) -> gateways
Line 118: are become
Line 165: are become
Line 173: in IoMT -> in the IoMT
Line 180: generated date of IoMT -> the IoMT
Line 181: IoMT -> the IoMT
Line 193: "no free lunch" is just an informal expression, not a formal theory, please correct accordingly.
Line 199: has been recently emerged
Line 266: difficult by -> difficult for?
Line 267: suspectable -> susceptible
Line 269: IoMT system facing -> the IoMT system is facing?
Line 294: in IoT -> in the IoT
Line 298: "works of using" doesn't read well
Line 302: of IoMT
Line 341: 500 papers were remained
Line 377: journal article -> journal articles
Line 379: were belong
Line 413: the true from false -> true from false
Line 427L the fake from -> fake from
Line 525: was failed -> failed
Line 571: requires -> require
Line 767/870: majority of -> the majority of
Line 850: who have -> who has
Line 858: articles that yet to be published - doesn't read well
Caption Table 2: different the IoMT -> different IoMT, reprsenst -> represents

Caption Fig.1: The entire Mind Map show The keywords used in the Research Information Template (RIT). The rectangle box at the middle represent -> shows/the keywords/rectangular/represents

I suggest adjusting the style of Fig. 7 to match Fig. 3 and Fig. 5. However, this is not required.

Reviewer 3 ·

Basic reporting

No comment

Experimental design

No comment

Validity of the findings

No comment

Additional comments

The authors' changes, in response to mine and others' reviews, do satisfy well the concerns in my initial review. Thanks for producing a significantly better article.